# TD Convergence: An Optimization Perspective

**Kavosh Asadi**[*]
Amazon

**Shoham Sabach**[*]
Amazon & Technion

**Yao Liu**
Amazon

**Omer Gottesman**
Amazon

**Rasool Fakoor**
Amazon

## Abstract

We study the convergence behavior of the celebrated temporal-difference (TD) learning algorithm. By looking at the algorithm through the lens of optimization, we first argue that TD can be viewed as an iterative optimization algorithm where the function to be minimized changes per iteration. By carefully investigating the divergence displayed by TD on a classical counter example, we identify two forces that determine the convergent or divergent behavior of the algorithm. We next formalize our discovery in the linear TD setting with quadratic loss and prove that convergence of TD hinges on the interplay between these two forces. We extend this optimization perspective to prove convergence of TD in a much broader setting than just linear approximation and squared loss. Our results provide a theoretical explanation for the successful application of TD in reinforcement learning.

## 1 Introduction

Temporal-difference (TD) learning is arguably one of the most important algorithms in reinforcement learning (RL), and many RL algorithms are based on principles that TD embodies. TD is at the epicenter of some of the recent success examples of RL [1, 2], and has influenced many areas of science such as AI, economics, and neuroscience. Despite the remarkable success of TD in numerous settings, the algorithm is shown to display divergent behavior in contrived examples [3, 4, 5]. In practice, however, divergence rarely manifests itself even in situations where TD is used in conjunction with complicated function approximators. Thus, it is worthwhile to obtain a deeper understanding of TD, and to generalize existing convergence results to explain its practical success.

In this paper, our desire is to study TD through the lens of optimization. We argue that TD could best be thought of as an iterative optimization algorithm, which proceeds as follows:

$$\theta^{t+1} \approx \arg\min_w H(\theta^t, w) . \tag{1}$$

Here, the objective $H$ is constructed by a) choosing a function approximator such as a neural network and b) defining a discrepancy measure between successive predictions, such as the squared error. We will discuss more examples later.

This process involves two different parameters, namely the target parameter $\theta^t$ that remains fixed at each iteration $t$, and the optimization parameter $w$ that is adjusted during each iteration to minimize the corresponding loss function. Using the more familiar deep-RL terminology, $\theta^t$ corresponds to the parameters of the target network, whereas $w$ corresponds to the parameters of the online network. Many RL algorithms can be described by this iterative process with the main difference being the approach taken to (approximately) perform the minimization. At one side of the spectrum lies the original TD algorithm [6], which proceeds by taking a single gradient-descent step to adjust $w$ and immediately updating $\theta$. More generally, at each iteration $t$ we can update the $w$ parameter $K$ times

---

[*]Equal contribution

37th Conference on Neural Information Processing Systems (NeurIPS 2023).

using gradient-descent while freezing $\theta$, a common technique in deep RL [1]. Finally, Fitted Value Iteration [7] lies at the other extreme and solves each iteration exactly when closed-form solutions exist. Therefore, a deeper understanding of the iterative optimization process (1) can facilitate a better understanding of TD and related RL algorithms.

The iterative process (1) is well-studied in the setting where $H$ is constructed using linear function approximation and squared error. In particular, with $K = 1$, meaning TD without frozen target network, the most notable result is due to the seminal work of Tsitsiklis and Van Roy [3] that proved the convergence of the algorithm. More recently, Lee and He [8] focused on the more general case of $K \geq 1$. While they took a major step in understanding the role of a frozen target-network in deep RL, they leaned on the more standard optimization tools to show that gradient-descent with a fixed $K$ can be viewed as solving each iteration approximately. Therefore, in their analysis, each iteration results in some error. This error is accumulated per iteration and needs to be accounted for in the final result. Therefore, Lee and He [8] fell short of showing convergence to the TD fixed-point, and managed to show that the final iterate will be in the vicinity of the fixed-point defined by the amount of error accumulated over the iterations.

In this work, our primary contribution is to generalize and improve existing results on TD convergence. In particular, in our theory, we can support the popular technique of employing frozen target networks with general $K \geq 1$, and we also support the use of a family of function approximators and error-measure pairs that includes linear function-approximation and squared error as a special case. To the best of our knowledge, this is the first contraction result that does not restrict to the case of $K = 1$, and can also generalize the well-explored linear case with quadratic error.

To build intuition, we start by taking a deeper dive into one of the classical examples where TD displays divergent behavior [3]. In doing so, we identify two key forces, namely a target force and an optimization force, whose interplay dictates whether TD is guaranteed to converge or that a divergent behavior may manifest itself. If the optimization force can dominate the target force, the process is convergent even when we operate in the famous deadly triad [5], namely in the presence of bootstrapping, function approximation, and off-policy updates. Overall, our results demonstrate that TD is a sound algorithm in a much broader setting than understood in previous work.

## 2 Problem Setting

Reinforcement learning (RL) is the study of artificial agents that can learn through trial and error [5]. In this paper, we focus on the setting where the agent is interested in predicting the long-term goodness or the value of its states. Referred to as the prediction setting, this problem is mathematically formulated by the Markov reward process (MRP) [9]. In this paper, we consider the discounted infinite-horizon case of MRPs, which is specified by the tuple $\langle \mathcal{S}, \mathcal{R}, \mathcal{P}, \gamma \rangle$, where $\mathcal{S}$ is the set of states. The function $R : \mathcal{S} \to \mathbb{R}$ denotes the reward when transitioning out of a state. For any set, we denote the space of probability distributions over it by $\Delta$. The transition $\mathcal{P} : \mathcal{S} \to \Delta(\mathcal{S})$ defines the conditional probabilities over the next states given the current state, and is denoted by $\mathcal{P}(s' \mid s)$. Finally, the scalar $\gamma \in (0, 1)$ geometrically discounts rewards that are received in the future steps.

The primary goal of this paper is to understand the behavior of RL algorithms that learn to approximate the state value function defined as $v(s) := \mathbb{E}\big[\sum_{t=0}^{\infty} \gamma^t r_t \big| s_0 = s\big]$. To this end, we define the Bellman operator $\mathcal{T}$ as follows:

$$\big[\mathcal{T}v\big](s) := \mathcal{R}(s) + \sum_{s' \in \mathcal{S}} \gamma \, \mathcal{P}(s' \mid s)v(s') \, ,$$

which we can write compactly as: $\mathcal{T}v := R + \gamma Pv$. In large-scale RL problems the number of states $|\mathcal{S}|$ is enormous, which makes it unfeasible to use tabular approaches. We focus on the setting where we have a parameterized function approximator, and our desire is to find a parameter $\theta$ for which the learned value function $v(s; \theta)$ results in a good approximation of the value function $v(s)$.

A fundamental and quite popular approach to finding a good approximation of the value function is known as temporal difference (TD) learning [6]. Suppose that a sample $\langle s, r, s' \rangle$ is given where $s' \sim \mathcal{P}(\cdot|s)$. In this case, TD learning algorithm updates the parameters of the approximate value function as follows:

$$\theta^{t+1} \leftarrow \theta^t + \alpha\big(r + \gamma v(s'; \theta^t) - v(s; \theta^t)\big)\nabla_\theta v(s; \theta^t) \, , \tag{2}$$

where $\theta^t$ denotes the parameters of our function approximator at iteration $t$. Also, by $\nabla_\theta v(s; \theta^t)$ we are denoting the partial gradient of $v(s; \theta)$ with respect to the parameter $\theta$. Note that TD uses

the value estimate obtained by one-step lookahead $\left(r + \gamma v(s'; \theta^t)\right)$ to update its approximate value function $v(s; \theta^t)$ in the previous step. This one-step look-ahead of TD could be thought of as a sample of the right-hand-side of the Bellman equation. Our focus on TD is due to the fact that many of the more modern RL algorithms are designed based on the principles that TD embodies. We explain this connection more comprehensively in section 3.

To understand the existing results on TD convergence, we define the Markov chain's stationary state distribution $d(\cdot)$ as the unique distribution with the following property: $\forall s' \sum_{s \in \mathcal{S}} d(s)\mathcal{P}(s' \mid s) = d(s')$. Then, Tsitsiklis and Van Roy show in [3] that, under linear function approximation, TD will be convergent if the states $s$ are sampled from the stationary-state distribution. We will discuss this result in more detail later on in section 5.

However, it is well-known that linear TD can display divergent behavior if states are sampled from an arbitrary distribution rather than the stationary-state distribution of the Markov chain. A simple yet classical counter example of divergence of linear TD is shown in Figure 1 and investigated in section 4. First identified in [3], this example is a very simple Markov chain with two non-terminal states and zero rewards. Moreover, little is known about convergence guarantees of TD under alternative function approximators, or even when the update rule of TD is modified slightly.

In this paper, we focus on the Markov reward process (MRP) setting which could be thought of as a Markov decision process (MDP) with a single action. TD can display divergence even in the MRP setting [3, 5], which indicates that the presence of multiple actions is not the core reason behind TD misbehavior [5] — Divergence can manifest itself even in the more specific case of single action. Our results can naturally be extended to multiple actions with off-policy updates where in update (2) of TD, actions are sampled according to a target policy but states are sampled according to a behavior policy. That said, akin to Tsitsiklis and Van Roy [3], as well as chapter 11 in the book of Sutton and Barto [5], we focus on MRPs to study the root cause of TD in the most clear setting.

## 3 TD Learning as Iterative Optimization

In this section, we argue that common approaches to value function prediction can be viewed as iterative optimization algorithms where the function to be minimized changes per iteration. To this end, we recall that for a given experience tuple $\langle s, r, s' \rangle$, TD performs the update presented in (2). A common augmentation of TD is to decouple the parameters to target ($\theta$) and optimization ($w$) parameters [1] and update $\theta$ less frequently. In this case, at a given iteration $t$, the algorithm performs multiple ($K$) gradient steps as follows:

$$w^{t,k+1} \leftarrow w^{t,k} + \alpha\left(r + \gamma v(s'; \theta^t) - v(s; w^{t,k})\right)\nabla_\theta v(s; w^{t,k}) , \tag{3}$$

and then updates the target parameter $\theta^{t+1} \leftarrow w^{t,K}$ before moving to the next iteration $t + 1$. Here $K$ is a hyper-parameter, where $K = 1$ takes us back to the original TD update (2). Observe that the dependence of $v(s'; \theta^t)$ to our optimization parameter $w$ is ignored in this update, despite the fact that an implicit dependence is present due to the final step $\theta^{t+1} \leftarrow w^{t,K}$. This means that the objective function being optimized is made up of two separate input variables[2]. We now define:

$$H(\theta, w) = \frac{1}{2} \sum_s d(s)\left(\mathbf{E}_{r,s'}[r + \gamma v(s'; \theta)] - v(s; w)\right)^2 ,$$

where we allow $d(\cdot)$ to be an arbitrary distribution, not just the stationary-state distribution of the Markov chain. Observe that the partial gradient of $H$ with respect to the optimization parameters $w$ is equivalent to the expectation of the update in (3). Therefore, TD, DQN, and similar algorithms could best be thought of as learning algorithms that proceed by approximately solving for the following sequence of optimization problems:

$$\theta^{t+1} \approx \arg\min_w H(\theta^t, w) ,$$

using first-order optimization techniques. This optimization perspective is useful conceptually because it accentuates the unusual property of this iterative process, namely that the first argument of the objective $H$ hinges on the output of the previous iteration. It also has important practical ramifications when designing RL optimizers [11]. Moreover, the general form of this optimization

---

[2]In fact, [10] shows that there cannot exist any objective function $J(\theta)$ with a single input variable whose gradient would take the form of the TD update.

process allows for using alternative forms of loss functions such as the Huber loss [1], the logistic loss [12], or the entropy [13], as well as various forms of function approximation such as linear functions or deep neural networks. Each combination of loss functions and function approximators yields a different $H$, but one that is always comprised of a function $H$ with two inputs.

A closely related optimization process is one where each iteration is solved exactly:

$$\theta^{t+1} \leftarrow \arg\min_w H(\theta^t, w) , \tag{4}$$

akin to Fitted Value Iteration [7, 14]. Exact optimization is doable in problems where the model of the environment is available and that the solution takes a closed form. A pseudo-code of both algorithms is presented in Algorithms 1 and 2.

---

**Algorithm 1** Value Function Optimization with Exact Updates

  **Input:** $\theta^0$, $T$
  **for** $t = 0$ **to** $T - 1$ **do**
    $\theta^{t+1} \leftarrow \arg\min_w H(\theta^t, w)$
  **end for**
  **Return** $\theta^T$

---

**Algorithm 2** Value Function Optimization with Gradient Updates

  **Input:** $\theta^0$, $T$, $K$, $\alpha$
  **for** $t = 0$ **to** $T - 1$ **do**
    $w^{t,0} \leftarrow \theta^t$
    **for** $k = 0$ **to** $K - 1$ **do**
      $w^{t,k+1} \leftarrow w^{t,k} - \alpha \nabla_w H(\theta^t, w^{t,k})$
    **end for**
    $\theta^{t+1} \leftarrow w^{t,K}$
  **end for**
  **Return** $\theta^T$

---

In terms of the difference between the two algorithms, notice that in Algorithm 1 we assume that we have the luxury of somehow solving each iteration exactly. This stands in contrast to Algorithm 2 where we may not have this luxury, and resort to gradient updates to find a rough approximation of the actual solution. Thus, Algorithm 1 is more difficult to apply but easier to understand, whereas Algorithm 2 is easier to apply but more involved in terms of obtaining a theoretical understanding.

Note that if convergence manifests itself in each of the two algorithms, the convergence point denoted by $\theta^\star$ must have the following property:

$$\nabla_w H(\theta^\star, \theta^\star) = \mathbf{0} . \tag{5}$$

This fixed-point characterization of TD has been explored in previous work [15, 16]. Whenever it exists, we refer to $\theta^\star$ as the fixed-point of these iterative algorithms. However, convergence to the fixed-point is not always guaranteed [5] even when we have the luxury of performing exact minimization akin to Algorithm 1. In this paper, we study both the exact and inexact version of the optimization process. In doing so, we identify two forces that primarily influence convergence. To begin our investigation, we study a counter example to build intuition on why divergence can manifest itself. We present a formal analysis of the convergence of the two algorithms in section 6.

## 4 Revisiting the Divergence Example of TD

In this section, we focus on a simple counter example where TD is known to exhibit divergence. First identified by [3], investigating this simple example enables us to build some intuition about the root cause of divergence in the most clear setting.

Shown in Figure 1, this example is a Markov chain with two non-terminal states and zero rewards. A linear function approximation, $v(s; \theta) = \phi(s)\theta$, is employed with a single feature where $\phi(s_1) = 1$ and $\phi(s_2) = 2$. The third state is a terminal one whose value is always zero. The true value function (0 in all states) is realizable with $\theta = 0$.

To build some intuition about the root cause of divergence, we discuss the convergence of exact TD with a few state distributions in this example. We desire to update all but the terminal state with non-zero probability. However, to begin with, we focus on a particular extreme case where we put all of our update probability behind the second state:

$$\theta^{t+1} \leftarrow \arg\min_w H(\theta^t, w) = \arg\min_w \frac{1}{2}\big((1 - \epsilon)(\gamma 2\theta^t) - 2w\big)^2 .$$

We thus have $\nabla_w H(\theta^t, w) = 2\big(2w - (1 - \epsilon)\gamma 2\theta^t\big)$, and since $\nabla_w H(\theta^t, \theta^{t+1}) = 0$, we can write: $\theta^{t+1} \leftarrow (2)^{-1}(1 - \epsilon)\gamma 2\theta^t$. The process converges to the fixed-point $\theta = 0$ for all values of $\gamma < 1$

and $\epsilon \in [0, 1]$. This is because the target force of $\theta_t$ (namely $((1 - \epsilon)\gamma 2)$) is always dominated by the optimization force of $w$ (namely 2). Note that updating this state is thus not problematic at all, and that the update becomes even more conducive to convergence when $\gamma$ is smaller and $\epsilon$ is larger.

We now juxtapose the first extreme case with the second one where we put all of our update probability behind the first state, in which case at a given iteration $t$ we have:

$$\theta^{t+1} \leftarrow \arg\min_w H(\theta^t, w) = \arg\min_w \frac{1}{2}(\gamma 2\theta^t - w)^2 \ .$$

We thus have $\nabla_w H(\theta^t, w) = (w - \gamma 2\theta^t)$ and so $\theta^{t+1} \leftarrow (1)^{-1}\gamma 2\theta^t$. Unlike the first extreme case, convergence is no longer guaranteed. Concretely, to ensure convergence, we require that the target force of $\theta^t$ (namely $\gamma 2$) be larger than the optimization force of $w$ (namely 1).

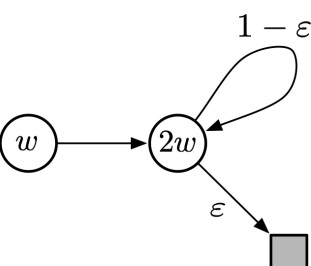

Figure 1: Divergence example of TD [3].

To better understand why divergence manifests itself, note that the two states $s_1$ and $s_2$ have very similar representations (in the form of a single feature). Note also that the value of the feature is larger for the target state than it is for the source state, $\phi(s_2) > \phi(s_1)$. This means that any change in the value of the source state $s_1$, would have the external effect of changing the value of the target state $s_2$. Further, the change is in the same direction (due to the positive sign of the feature in both states) and that it is exactly $2\gamma$ larger for the target state relative to the source state. Therefore, when $\gamma > 1/2$, the function $H$ will be more sensitive to the target parameter $\theta$ relative to the optimization parameter $w$. This is the root cause of divergence.

Moving to the case where we update both states with non-zero probability, first note that if the two updates were individually convergent, then the combination of the two updates would also have been convergent in a probability-agnostic way. Stated differently, convergence would have been guaranteed regardless of the probability distribution $d$ and for all off-policy updates. However, in light of the fact that the optimization force of $s_1$ does not dominate its target force, we need to choose $d(s_1)$ small enough so as to contain the harmful effect of updating this state.

We compute the overall update (under the case where we update the two states with equal probability) by computing the sum of the two gradients in the two extreme cases above:

$$(w - \gamma 2\theta^t) + 2(2w - \gamma 2(1 - \epsilon)\theta^t) = 0 \ ,$$

and so $\theta^{t+1} \leftarrow (5)^{-1}\gamma(6 - 4\epsilon)\theta^t$. Note that even with a uniform update distribution, the two states are contributing non-uniformly to the overall update, and the update in the state $s_2$ is more influential because of the higher magnitude of the feature in this state ($\phi(s_2) = 2$ against $\phi(s_1) = 1$).

To ensure convergence, we need to have that $\gamma < \frac{5}{6 - 4\epsilon}$. Notice, again, that we are combining the update in a problematic state with the update in the state that is conducive for convergence. We can contain the negative effect of updating the first state by ensuring that our update in the second state is even more conducive for convergence (corresponding to a larger $\epsilon$). In this case, the update of $s_2$ can serve as a mitigating factor.

We can further characterize the update with a general distribution. In this case, we have:

$$d(s_1)\big((w - \gamma 2\theta^t)\big) + \big(1 - d(s_1)\big)\big(2(2w - \gamma 2(1 - \epsilon)\theta^t)\big) = 0 \ ,$$

which gives us a convergent update if: $d(s_1) < \frac{4 - 4\gamma(1-\epsilon)}{3 + \gamma 2 - 4\gamma(1-\epsilon)}$ . Both the denominator and the numerator are always positive, therefore regardless of the values of $\epsilon$ and $\gamma$, there always exists a convergent off-policy update, but one that needs to assign less probability to the problematic state as we increase $\gamma$ and decrease $\epsilon$.

This means that using some carefully chosen off-policy distributions is not only safe, but that doing so can even speed up convergence. This will be the case if the chosen distribution makes the objective function $H$ more sensitive to changes in its first argument (target parameter $\theta$) than its second argument (the optimization parameter $w$). We next formalize this intuition.

# 5 Convergence of Linear TD with Quadratic Loss

We now focus on the case of TD with linear function approximation. In this case, the expected TD update (2) could be written as the composition of two separate operators (with $v_t = \Phi\theta_t$):

$$v_{t+1} \leftarrow \Pi_D\big(\mathcal{T}(v_t)\big) \,,$$

where the projection operator $\Pi_D$ and the Bellman operator $\mathcal{T}$ are defined as follows:

$$\Pi_D = \Phi(\Phi^\top D\Phi)^{-1}\Phi^\top D, \qquad \text{and} \qquad \mathcal{T}(v) = R + \gamma Pv \,.$$

Here, $D$ is a diagonal matrix with diagonal entries $d(s_1), ..., d(s_n)$, where $n$ is the number of states. The projection operator $\Pi_D$ is non-expansive under any distribution $d$. However, the Bellman operator is a $\gamma$-contraction under a specific $d$, namely the stationary-state distribution of the Markov chain specified by the transition matrix $P$ as shown by Tsitsiklis and Van Roy [3]. Therefore, the composition of the two operators is a $\gamma$-contraction when the distribution $d$ is the stationary-state distribution of the Markov chain.

In light of this result, one may think that TD is convergent in a very narrow sense, specifically when the updates are performed using the stationary-state distribution. But, as we saw with the counter example, in general there may exist many other distributions $d$ that are in fact very conducive for TD convergence. In these cases, the proof technique above cannot provide a tool to ensure convergence because of the reliance of the non-expansive and contraction properties of these operators. Is this a limitation of the TD algorithm itself, or a limitation of this specific operator perspective of TD? Further, if this operator perspective is limited, is there a different perspective that can give rise to a more general understanding of TD convergence?

Our goal for the rest of the paper is to develop an alternative optimization perspective that can be applied in a broader setting relative to the operator perspective. To this end, our first task is to show that the optimization perspective can give us new insights even in the linear case and with squared loss functions. We generalize our optimization view of TD in the counter example by defining the objective function $H$ for this case:

$$H(\theta, w) = \frac{1}{2}\|R + \gamma P\Phi\theta - \Phi w\|_D^2 \,, \tag{6}$$

where $\|x\|_D = \sqrt{x^\top Dx}$ . Recall that in the exact case, the TD algorithm could succinctly be written as: $\theta^{t+1} \leftarrow \arg\min_w H(\theta^t, w)$. Following the steps taken with the counter example, we first compute the gradient to derive the target and optimization forces:

$$\nabla_w H(\theta, w) = \Phi^\top D(\Phi w - R - \gamma P\Phi\theta) = \underbrace{\Phi^\top D\Phi}_{\mathbf{M}_w} w - \underbrace{\gamma\Phi^\top DP\Phi}_{\mathbf{M}_\theta}\theta - \Phi^\top DR \,. \tag{7}$$

Here we have similar dynamics between $\theta$ and $w$ except that in this more general case, rather than scalar forces as in the counter example, we now have the two matrices $\mathbf{M}_w$ and $\mathbf{M}_\theta$. Note that $\mathbf{M}_w$ is a positive definite matrix, $\lambda_{\min}(\mathbf{M}_w) = \min_x x^\top \mathbf{M}_w x/\|x\|^2 > 0$, if $\Phi$ is full rank. Below we can conveniently derive the update rule of linear TD by using these two matrices. All proofs are in the appendix.

**Proposition 1.** Let $\{\theta^t\}_{t\in\mathbb{N}}$ be a sequence of parameters generated by Algorithm 1. Then, for the fixed-point $\theta^\star$ and any $t \in \mathbb{N}$, we have

$$\theta^{t+1} - \theta^\star = \mathbf{M}_w^{-1}\mathbf{M}_\theta\left(\theta^t - \theta^\star\right).$$

This proposition characterizes the evolution of the difference between the parameter $\theta^t$ and the fixed-point $\theta^\star$. Similarly to our approach with the counter example, we desire to ensure that this difference converges to $\mathbf{0}$. The following corollary gives us a condition for convergence [17].

**Corollary 2.** Let $\{\theta^t\}_{t\in\mathbb{N}}$ be a sequence of parameters generated by Algorithm 1. Then, $\{\theta^t\}_{t\in\mathbb{N}}$ converges to the fixed-point $\theta^\star$ if and only if the spectral radius of $\mathbf{M}_w^{-1}\mathbf{M}_\theta$ satisfies $\rho(\mathbf{M}_w^{-1}\mathbf{M}_\theta) < 1$.

We can employ this Corollary to characterize the convergence of TD in the counter example from the previous section. In this case, we have $\mathbf{M}_w = 5$ and $\mathbf{M}_\theta = \gamma(6 - 4\epsilon)$, which give us $(5)^{-1}\gamma(6 - 4\epsilon) < 1$. This is exactly the condition obtained in the previous section.

Notice that if $d$ is the stationary-state distribution of the Markov chain then $\rho(\mathbf{M}_w^{-1}\mathbf{M}_\theta) < 1$ [3], and so the algorithm is convergent. However, as demonstrated in the counter example, the condition can also hold for many other distributions. The key insight is to ensure that the distribution puts more weight behind states where the optimization force of $w$ is dominating the target force due to $\theta$.

Note that the operator view of TD becomes inapplicable as we make modifications to $H$, because it will be unclear how to write the corresponding RL algorithm as a composition of operators. Can we demonstrate that in these cases the optimization perspective is still well-equipped to provide us with new insights about TD convergence? We next answer this question affirmatively by showing that the optimization perspective of TD convergence naturally extends to a broader setting than considered here, and therefore, is a more powerful perspective than the classical operator perspective.

## 6 Convergence of TD with General $H$

Our desire now is to show convergence of TD without limiting the scope of our results to linear approximation and squared loss. We would like our theoretical results to support alternative ways of constructing $H$ than the one studied in existing work as well as our previous section. In doing so, we again show that convergence of TD hinges on the interplay between the two identified forces.

Before presenting our main theorem, we discuss important concepts from optimization that will be used at the core of our proofs. We study convergence of Algorithms 1 and 2 with a general function $H : \mathbb{R}^n \times \mathbb{R}^n \to \mathbb{R}$ that satisfies the following two assumptions:

    I. The partial gradient $\nabla_w H$, is $F_\theta$-Lipschitz:

$$\forall \theta_1, \forall \theta_2 \qquad \|\nabla_w H(\theta_1, w) - \nabla_w H(\theta_2, w)\| \leq F_\theta \|\theta_1 - \theta_2\| .$$

    II. The function $H(\theta, w)$ is $F_w$-strongly convex in $w$:

$$\forall w_1, \forall w_2 \qquad \left(\nabla_w H(\theta, w_1) - \nabla_w H(\theta, w_2)\right)^\top (w_1 - w_2) \geq F_w \|w_1 - w_2\|^2 .$$

Note that, in the specific linear case and quadratic loss (our setting in the previous section) these assumptions are easily satisfied [8]. More specifically, in that case $F_\theta = \lambda_{max}(\mathbf{M}_\theta)$ and $F_w = \lambda_{min}(\mathbf{M}_w)$. But the assumptions are also satisfied in a much broader setting than before. We provide more examples in section 6.3. We are now ready to present the main result of our paper:

**Theorem 3.** Let $\{\theta^t\}_{t\in\mathbb{N}}$ be a sequence generated by either Algorithm 1 or 2. If $F_\theta < F_w$, then the sequence $\{\theta^t\}_{t\in\mathbb{N}}$ converges to the fixed-point $\theta^\star$.

In order to prove the result, we tackle the two cases of Algorithm 1 and 2 separately. We first start by showing the result for Algorithm 1 where we can solve each iteration exactly. This is an easier case to tackle because we have the luxury of performing exact minimization, which is more stringent and difficult to implement but easier to analyze and understand. This would be more pertinent to Fitted Value Iteration and similar algorithms. We then move to the case where we approximately solve each iteration (Algorithm 2) akin to TD and similar algorithms. The proof in this case is more involved, and partly relies on choosing small steps when performing gradient descent.

### 6.1 Exact Optimization (Algorithm 1)

In this case, convergence to the fixed-point $\theta^\star$ can be obtained as a corollary of the following result:

**Proposition 4.** Let $\{\theta^t\}_{t\in\mathbb{N}}$ be a sequence generated by Algorithm 1. Then, we have:

$$\|\theta^{t+1} - \theta^\star\| \leq F_w^{-1} F_\theta \|\theta^t - \theta^\star\| .$$

From this result, we immediately obtain that the relative strength of the two forces, namely the conducive force $F_w$ due to optimization and the detrimental target force $F_\theta$, determines if the algorithm is well-behaved. The proof of Theorem 3 in this case follows immediately from Proposition 4.

### 6.2 Inexact Optimization (Algorithm 2)

So far we have shown that the optimization process is convergent in the presence of exact optimization. This result supports the soundness of algorithms such as Fitted Value Iteration, but not TD yet, because in the case of TD we only roughly approximate the minimization step. Can this desirable

convergence result be extended to the more general setting of TD-like algorithms where we inexactly solve each iteration by a few gradient steps, or is exact optimization necessary for obtaining convergence? Answering this question is very important because in many settings it would be a stringent requirement to have to solve the optimization problem exactly.

We now show that indeed convergence manifests itself in the inexact case as well. In the extreme case, we can show that all we need is merely one single gradient update at each iteration. This means that even the purely online TD algorithm, presented in update (2), is convergent with general $H$ if the optimization force can dominate the target force.

However, it is important to note that because we are now using gradient information to crudely approximate each iteration, we need to ensure that the step-size parameter $\alpha$ is chosen reasonably. More concretely, in this setting we need an additional assumption, namely that there exists an $L > 0$ such that:

$$\forall w_1, \forall w_2 \qquad \|\nabla_w H(\theta, w_1) - \nabla_w H(\theta, w_2)\| \leq L\|w_1 - w_2\| \, .$$

Notice that such an assumption is quite common in the optimization literature (see, for instance, [18]). Moreover, it is common to choose $\alpha = 1/L$, which we also employ in the context of Algorithm 2. We formalize this in the proposition presented below:

**Proposition 5.** Let $\{\theta^t\}_{t\in\mathbb{N}}$ be a sequence generated by Algorithm 2 with the step-size $\alpha = 1/L$. Then, we have:

$$\|\theta^{t+1} - \theta^\star\| \leq \sigma_K \|\theta^t - \theta^\star\| \, ,$$

where:

$$\sigma_K^2 \equiv (1 - \kappa)^K \left(1 - \eta^2\right) + \eta^2 \, .$$

with $\kappa \equiv L^{-1}F_w$ and $\eta \equiv F_w^{-1}F_\theta$.

Notice that $\kappa$, which is sometimes referred to as the inverse condition number in the optimization literature, is always smaller than 1. Therefore, we immediately conclude Theorem 3. Indeed, since the optimization force dominates the target force (meaning $\eta < 1$), Algorithm 2 is convergent. Notice that a surprisingly positive consequence of this theorem is that we get convergent updates even if we only perform one gradient step per iteration ($K = 1$). In deep-RL terminology, this corresponds to the case where we basically have no frozen target network, and that we immediately use the new target parameter $\theta$ for the subsequent update.

To further situate this result, notice that as $K$ approaches $\infty$ then $\sigma_K$ approaches $\eta$, which is exactly the contraction factor from Proposition 4 where we assumed exact optimization. So this proposition should be thought of as a tight generalization of Proposition 4 for the exact case. With a finite $K$ we are paying a price for the crudeness of our approximation.

Moreover, another interesting reduction of our result is to the case where the target force due to bootstrapping in RL is absent, meaning that $F_\theta \equiv 0$. In this case, the contraction factor $\sigma_K$ reduces to $(1 - \kappa)^{K/2}$, which is exactly the known convergence rate for the gradient-descent algorithm in the strongly convex setting [18].

### 6.3 Examples

We focus on two families of loss functions where our assumptions can hold easily. To explain the first family, recall that TD could be written as follows:

$$\theta^{t+1} \leftarrow \arg\min_w H(\theta^t, w).$$

Now suppose we can write the function $H(\theta, w)$ as the sum of two separate functions $H(\theta, w) = G(\theta, w) + L(w)$, where the function $L(w)$ is strongly convex with respect to $w$. This setting is akin to using ridge regularization [19], which is quite common in deep learning (for example, the very popular optimizer AdamW [20]). This allows us to now work with functions $G$ that are only convex (in fact, weakly convex is enough) with respect to $w$. We provide two examples:

- Suppose we would like to stick with linear function approximation. Then, the function $G$ could be constructed using any convex loss where $\nabla_w G(\theta, w)$ is Lipschitz-continuous with respect to $\theta$. Examples that satisfy this include the Huber loss [1] or the Logistic loss [12].

- Suppose we want to use the more powerful convex neural networks [21]. We need the error to be convex and monotonically increasing so that $G$ is still convex. This is due to the classical result on the composition of convex functions. One example is the quadratic error where we restrict the output of the function approximator to positive values. Such neural nets are also Lipschitz continuous given proper activation functions such as ReLU.

Beyond this family, we have identified a second family, namely the control setting where a greedification operator is needed for bootstrapping. For example, with the quadratic error we could have:

$$H(\theta, w) = \frac{1}{2} \sum_s d(s) \sum_a \pi(a|s)(\mathbb{E}_{s'}[r + \gamma \max_{a'} q(s', a', \theta)] - q(s, a, w))^2,$$

where $q$ is the state-action value function parameterized by either $\theta$ or $w$.

We again need the two assumptions, namely strong-convexity with respect to $w$ and Lipschitzness of $\nabla_w H(\theta, w)$ with respect to $\theta$ to hold. Actually, Lee and He [8] already showed the strong convexity with respect to $w$, but we need to still show the Lipschitz property of $\nabla_w H(\theta, w)$ with respect to $\theta$. Note that they showed the Lipschitz property only with respect to $w$ and not with respect to $\theta$. We are now able to show this result. Please see Proposition 8 and its proof in the appendix. Our proof also supports other greedification operators, such as softmax [22], so long as its non-expansive.

## 7    Related Work

In this paper, we studied convergence of TD through the lens of optimization. The underlying principles of TD are so central to RL that a large number of RL algorithms can be thought of as versions of TD, and the availability of convergence results varies between different types of algorithms. We chose a setting we believe is as elementary as possible to highlight key principles of TD convergence. Closest to our work is the work of Tsitsiklis and Van Roy [3] who proved convergence of TD with linear function approximation, squared error, and the stationary-state distribution of the Markov chain. Later work, especially that of Bhandari et al. [23], further analyzed the case of TD with stochastic gradients and non-iid samples in the on-policy case. Extensions to the control setting also exist [24, 25].

In particular, Melo et al. [24] presented a condition in their equation (7) which may look similar to our condition at first glance, but it is in fact quite different than the condition identified in our paper. In their equation (7), they require that the eigenvalues of $\Sigma_\pi$, the policy-conditioned covariance matrix $\Phi^\top D \Phi$, dominate the eigenvalues of a second matrix $\gamma^2 \Sigma_\pi^\star(\theta)$. Here $\Sigma_\pi^\star(\theta)$ is a similar covariance matrix for features, but one that is computed based on the action-greedification step.

We believe to be the first paper to show a exact contraction for TD with frozen target network and general $K$. To the best of our knowledge, existing results prior to Lee and He [8] mainly considered the case where we either never freeze the target network (corresponding to the value of $K = 1$), or the somewhat unrealistic case where we can exactly solve each iteration. Lee and He showed guarantees pertaining to $K > 1$, but notice that, while their result is quite innovative, they leaned on the more standard optimization tools for ensuring that gradient descent with a fixed $K$ can only solve each iteration approximately. Therefore, each iteration results in some error. In their theory, this error is accumulated per iteration and needs to be accounted for in the final result. Therefore, they fell short of showing 1) contraction and 2) exact convergence to the TD fixed-point, and only show that the final iterate is in the vicinity of the fixed-point defined by the amount of error accumulated over the trajectory. With finite $K$, they need to account for errors in solving each iteration (denoted by $\epsilon_k$ in their proofs such as in Theorem 3), which prevent them from obtaining a clean convergence result. In contrast, we can show that even approximately solving each iteration (corresponding to finite $K$) is enough to obtain contraction (without any error), because we look at the net effect of $K$ updates to the online network and the single update to the target network, and show that this effect is on that always take us closer to the fixed-point irregardless of the specific value of $K$.

Modifications of TD are presented in prior work that are more conducive to convergence analysis [26, 27, 28, 29, 30]. They have had varying degrees of success both in terms of empirical performance [31], and in terms of producing convergent algorithms [32]. TD is also studied under over-parameterization [33, 34], with learned representations [35], proximal updates [36], and auxiliary tasks [37]. Also, the quality of TD fixed-point has been studied in previous work [16].

A large body of literature focuses on finding TD-like approaches that are in fact true gradient-descent approaches in that they follow the gradient of a stationary objective function [38, 15, 39, 40, 41, 42].

In these works, the optimization problem is formulated in such a way that the minimizer of the loss will be the fixed-point of the standard TD. Whereas TD has been extended to large-scale settings, these algorithms have not been as successful as TD in terms of applications.

A closely related algorithm to TD is that of Baird, namely the residual gradient algorithm [4, 43]. This algorithm has a double-sampling issue that needs to be addressed either by assuming a model of the environment or by learning the variance of the value function [43, 44]. However, even with deterministic MDPs, in which the double sampling issue is not present [45], the algorithm often finds a fixed-point that has a lower quality than that of the TD algorithm [46]. This is attributed to the fact that the MDP might still look stochastic in light of the use of function approximation [38].

TD could be thought of as an incremental approach to approximate dynamic programming and Fitted Value Iteration [7] for which various convergence results based on the operator view exits [14, 47, 48]. Also, these algorithm are well-studied in terms of their error-propagation behavior [49, 50, 51].

Many asymptotic or finite-sample results on Q-learning (the control version of TD) with function approximation make additional assumptions on the problem structure, with a focus on the exploration problems [52, 53, 54, 55, 56]. Like mentioned before, our focus was on the prediction setting where exploration is not relevant.

## 8 Open Questions

To foster further progress in this area we identify key questions that still remain open.

I. We showed that TD converges to the fixed-point $\theta^\star$ characterized by $\nabla_w H(\theta^\star, \theta^\star) = \mathbf{0}$ and so the sequence $\{\|\nabla_w H(\theta^t, \theta^t)\|\}_{t \in \mathbb{N}}$ converges to 0. Can we further show that this sequence monotonically decreases with $t$, namely that $\|\|\nabla_w H(\theta^{t+1}, \theta^{t+1})\| \leq \|\nabla_w H(\theta^t, \theta^t)\|$ for all $t \in \mathbb{N}$? Interestingly, a line of research focused on inventing new algorithms (often called gradient TD algorithms) that possess this property [57]. However, if true, this result would entail that when our assumptions hold the original TD algorithm gives us this desirable property for free.

II. Going back to the counter example, we can show convergent updates by constraining the representation as follows: $\phi(s) \leq \gamma \phi(s')$ whenever $\mathcal{P}(s' \mid s) > 0$. This result can easily be extended to all MDPs with a single-dimensional feature vector. A natural question then is to identify a multi-dimensional representation, given a fixed MDP and update distribution $D$, leads into convergent updates. Similarly, we can pose this question in terms of fixing the MDP and the representation matrix $\Phi$, and identifying a set of distributions that lead into convergent updates. Again, we saw from the counter example that putting more weight behind states whose target force is weak (such as states leading into the terminal state) is conducive to convergence. How do we identify such distributions systematically?

III. We studied the setting of deterministic updates. To bridge the gap with RL practice, we need to study TD under stochastic-gradient updates. The stochasticity could for instance be due to using a minibatch or using non-iid samples. Bhandari et al. [23] tackle this for the case of $K = 1$. Can we generalize our convergence result to this more realistic setting?

IV. We have characterized the optimization force using the notion of strong convexity. It would be interesting to relax this assumption to handle neural networks and alternative loss functions. This can serve as a more grounded explanation for the empirical success of TD in the context of deep RL. Can we generalize our result to this more challenging setting?

## 9 Conclusion

In this paper, we argued that the optimization perspective of TD is more powerful than the well-explored operator perspective of TD. To demonstrate this, we generalized previous convergence results of TD beyond the linear setting and squared loss functions. We believe that further exploring this optimization perspective can be a promising direction to design convergent RL algorithms.

Our general result on the convergent nature of TD is consistent with the empirical success and the attention that this algorithm has deservedly received. The key factor that governs the convergence of TD is to ensure that the optimization force of the algorithm is well-equipped to dominate the more harmful target force. This analogy is one that can be employed to explain the convergent nature of TD even in the presence of the three pillar of the deadly triad.

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

## 10 Appendix

**Proposition 1.** Let $\{\theta^t\}_{t\in\mathbb{N}}$ be a sequence of parameters generated by Algorithm 1. Then, for the fixed-point $\theta^\star$ and any $t \in \mathbb{N}$, we have

$$\theta^{t+1} - \theta^\star = \mathbf{M}_w^{-1}\mathbf{M}_\theta\left(\theta^t - \theta^\star\right).$$

*Proof.* Since $\theta^{t+1} \leftarrow \arg\min_w H(\theta^t, w)$, we have $\nabla_w H(\theta^t, \theta^{t+1}) = \mathbf{0}$. Using (7) we have:

$$\mathbf{M}_w\theta^{t+1} = \Phi^\top DR + \mathbf{M}_\theta\theta^t = (\mathbf{M}_w - \mathbf{M}_\theta)\theta^\star + \mathbf{M}_\theta\theta^t,$$

where the last equality follows from $\theta^\star$ being the fixed-point and therefore $\nabla_w h(\theta^\star, \theta^\star) = \mathbf{0}$, which in light of (7) translates to $\Phi^\top DR = (\mathbf{M}_w - \mathbf{M}_\theta)\theta^\star$. Multiplying both sides by $\mathbf{M}_w^{-1}$, we get:

$$\theta^{t+1} = (I - \mathbf{M}_w^{-1}\mathbf{M}_\theta)\theta^\star + \mathbf{M}_w^{-1}\mathbf{M}_\theta\theta^t.$$

Rearranging the equality leads into the desired result. $\qquad\square$

**Proposition 4.** Let $\{\theta^t\}_{t\in\mathbb{N}}$ be a sequence generated by Algorithm 1. Then, we have:

$$\|\theta^{t+1} - \theta^\star\| \leq F_w^{-1}F_\theta\|\theta^t - \theta^\star\|.$$

*Proof.* First notice that $\theta^{t+1} \leftarrow \arg\min_w H(\theta^t, w)$, so we have $\nabla_w H(\theta^t, \theta^{t+1}) = \mathbf{0}$. Now, using the $F_w$-strong convexity of $w \to H(\theta^t, w)$, we get that

$$
\begin{aligned}
F_w\|\theta^{t+1} - \theta^\star\|^2 &\leq \langle \theta^\star - \theta^{t+1}, \nabla_w H(\theta^t, \theta^\star) - \nabla_w H(\theta^t, \theta^{t+1})\rangle \\
&= \langle \theta^\star - \theta^{t+1}, \nabla_w H(\theta^t, \theta^\star)\rangle \qquad \text{(from } \nabla_w H(\theta^t, \theta^{t+1}) = \mathbf{0}).
\end{aligned}
$$

Now, since $\theta^\star$ is a fixed-point, it follows that $\nabla_w H(\theta^\star, \theta^\star) = \mathbf{0}$. Therefore, we have:

$$
\begin{aligned}
F_w\|\theta^{t+1} - \theta^\star\|^2 &\leq \langle \theta^\star - \theta^{t+1}, \nabla_w H(\theta^t, \theta^\star) - \nabla_w H(\theta^\star, \theta^\star)\rangle \\
&\leq \|\theta^{t+1} - \theta^\star\| \cdot \|\nabla_w H(\theta^t, \theta^\star) - \nabla_w H(\theta^\star, \theta^\star)\| \\
&\leq F_\theta\|\theta^{t+1} - \theta^\star\| \cdot \|\theta^t - \theta^\star\|,
\end{aligned}
$$

where in the second line we used to Cauchy-Schwartz inequality, and the last inequality follows from the $F_\theta$-Lipschitz property of $\nabla_w H(\cdot, \theta^\star)$. Since this inequality holds true for any $t \in \mathbb{N}$, we get that if $\theta^t = \theta^\star$, then we also have that $\theta^{t+1} = \theta^\star$. Thus, if $\theta^T = \theta^\star$ for some $T \in \mathbb{N}$, then $\theta^t = \theta^\star$ for any $t \geq T$ and so the algorithm has converged. On the other hand, if $\theta^t \neq \theta^\star$ for all $t \in \mathbb{N}$, the desired result follows after dividing both sides by $\|\theta^{t+1} - \theta^\star\|$. $\qquad\square$

To prove Proposition 5 we need to provide two new Propositions to later make use of. We present these two proofs next, and then restate and prove Proposition 5.

**Proposition 6.** Let $\{\theta^t\}_{t\in\mathbb{N}}$ be a sequence generated by Algorithm 2. Then, for all $t \in \mathbb{N}$ and $0 \le k \le K - 1$, we have

$$H(\theta^t, \theta^\star) - H(\theta^t, w^{t,k}) \le \frac{F_\theta^2}{2F_w}\|\theta^t - \theta^\star\|^2 - \left(\frac{1}{\alpha} - \frac{L}{2}\right)\|w^{t,k+1} - w^{t,k}\|^2.$$

*Proof.* Let $t \in \mathbb{N}$. From the $F_w$-strong convexity of $w \to H(\theta^t, w)$, we obtain that

$$H(\theta^t, w^{t,k+1}) \ge H(\theta^t, \theta^\star) + \langle \nabla_w H(\theta^t, \theta^\star), w^{t,k+1} - \theta^\star\rangle + \frac{F_w}{2}\|w^{t,k+1} - \theta^\star\|^2,$$

which means that

$$H(\theta^t, \theta^\star) - H(\theta^t, w^{t,k+1}) \le \langle \nabla_w H(\theta^t, \theta^\star), \theta^\star - w^{t,k+1}\rangle - \frac{F_w}{2}\|w^{t,k+1} - \theta^\star\|^2. \qquad (8)$$

Moreover, in light of the fixed-point characterization of TD, namely that $\nabla_w H(\theta^\star, \theta^\star) = \mathbf{0}$, we can write

$$\begin{aligned}
\langle \nabla_w H(\theta^t, \theta^\star), \theta^\star - w^{t,k+1}\rangle &= \langle \nabla_w H(\theta^t, \theta^\star) - \nabla_w H(\theta^\star, \theta^\star), \theta^\star - w^{t,k+1}\rangle \\
&= \left(\nabla_w H(\theta^t, \theta^\star) - \nabla_w H(\theta^\star, \theta^\star)\right)^\top \left(\theta^\star - w^{t,k+1}\right) \\
&\le \frac{1}{2F_w}\|\nabla_w H(\theta^t, \theta^\star) - \nabla_w H(\theta^\star, \theta^\star)\|^2 + \frac{F_w}{2}\|w^{t,k+1} - \theta^\star\|^2 \\
&\le \frac{F_\theta^2}{2F_w}\|\theta^t - \theta^\star\|^2 + \frac{F_w}{2}\|w^{t,k+1} - \theta^\star\|^2. \qquad (9)
\end{aligned}$$

Here, the first inequality follows from the fact that for any two vectors $a$ and $b$ we have $a^\top b \le (1/2d)\|a\|^2 + (d/2)\|b\|^2$ for any $d > 0$. In this case, we chose $d = F_w$. Also the last inequality follows from the $F_\theta$-Lipschitz property of $\nabla_w H$, which is our assumption.

Now, by combining (8) with (9) we obtain that

$$\begin{aligned}
H(\theta^t, \theta^\star) - H(\theta^t, w^{t,k+1}) &\le \frac{F_\theta^2}{2F_w}\|\theta^t - \theta^\star\|^2 + \frac{F_w}{2}\|w^{t,k+1} - \theta^\star\|^2 - \frac{F_w}{2}\|w^{t,k+1} - \theta^\star\|^2 \\
&= \frac{F_\theta^2}{2F_w}\|\theta^t - \theta^\star\|^2. \qquad (10)
\end{aligned}$$

From the Lipschitz assumption we can write, due to the Descent Lemma [18] applied to the function $w \to H(\theta^t, w)$, that:

$$H(\theta^t, w^{t,k+1}) - H(\theta^t, w^{t,k}) \le \langle \nabla_w H(\theta^t, w^{t,k}), w^{t,k+1} - w^{t,k}\rangle + \frac{L}{2}\|w^{t,k+1} - w^{t,k}\|^2$$

Now, notice that according to Algorithm 2 we have $w^{t,k+1} = w^{t,k} - \alpha\nabla_w H(\theta^t, w^{t,k})$, and so we can write:

$$\begin{aligned}
H(\theta^t, w^{t,k+1}) - H(\theta^t, w^{t,k}) &\le \frac{1}{\alpha}\langle w^{t,k} - w^{t,k+1}, w^{t,k+1} - w^{t,k}\rangle + \frac{L}{2}\|w^{t,k+1} - w^{t,k}\|^2 \\
&= -\left(\frac{1}{\alpha} - \frac{L}{2}\right)\|w^{t,k+1} - w^{t,k}\|^2. \qquad (11)
\end{aligned}$$

Adding both sides of (10) with (11) yields:

$$H(\theta^t, \theta^\star) - H(\theta^t, w^{t,k}) \le \frac{F_\theta^2}{2F_w}\|\theta^t - \theta^\star\|^2 - \left(\frac{1}{\alpha} - \frac{L}{2}\right)\|w^{t,k+1} - w^{t,k}\|^2,$$

which proves the desired result. $\qquad \square$

Now, we can prove the following result.

**Proposition 7.** Let $\{\theta^t\}_{t \in \mathbb{N}}$ be a sequence generated by Algorithm 2. Then, for all $t \in \mathbb{N}$ and $0 \le k \le K - 1$, we have

$$\|w^{t,k+1} - \theta^\star\|^2 \le (1 - \alpha F_w) \|w^{t,k} - \theta^\star\|^2 + \frac{\alpha F_\theta^2}{F_w} \|\theta^t - \theta^\star\|^2 - (2 - \alpha L) \|w^{t,k+1} - w^{t,k}\|^2.$$

In particular, when $\alpha = 1/L$, we have

$$\|w^{t,k+1} - \theta^\star\|^2 \le \left(1 - \frac{F_w}{L}\right) \|w^{t,k} - \theta^\star\|^2 + \frac{F_\theta^2}{LF_w} \|\theta^t - \theta^\star\|^2.$$

*Proof.* Let $t \in \mathbb{N}$. From the definition of steps of Algorithm 2, that is, $w^{t,k+1} = w^{t,k} - \alpha \nabla_w H(\theta^t, w^{t,k})$, for any $0 \le k \le K - 1$, we obtain that

$$\|w^{t,k+1} - \theta^\star\|^2 = \|(w^{t,k} - \theta^*) - \alpha \nabla_w H(\theta^t, w^{t,k})\|^2$$
$$= \|w^{t,k} - \theta^\star\|^2 + 2\alpha \langle \nabla_w H(\theta^t, w^{t,k}), \theta^\star - w^{t,k}\rangle + \|\alpha \nabla_w H(\theta^t, w^{t,k})\|^2$$
$$= \|w^{t,k} - \theta^\star\|^2 + 2\alpha \langle \nabla_w H(\theta^t, w^{t,k}), \theta^\star - w^{t,k}\rangle + \|w^{t,k+1} - w^{t,k}\|^2. \quad (12)$$

Using the $F_w$-strong convexity of $w \to H(\theta^t, w)$, we have

$$H(\theta^t, \theta^\star) \ge H(\theta^t, w^{t,k}) + \langle \nabla_w H(\theta^t, w^{t,k}), \theta^\star - w^{t,k}\rangle + \frac{F_w}{2} \|w^{t,k} - \theta^\star\|^2. \quad (13)$$

Combining (12) and (13), we get:

$$\|w^{t,k+1} - \theta^\star\|^2 \le \|w^{t,k} - \theta^\star\|^2 + 2\alpha \left( H(\theta^t, \theta^\star) - H(\theta^t, w^{t,k}) - \frac{F_w}{2} \|w^{t,k} - \theta^\star\|^2\right)$$
$$+ \|w^{t,k+1} - w^{t,k}\|^2$$
$$= (1 - \alpha F_w) \|w^{t,k} - \theta^\star\|^2 + 2\alpha \left( H(\theta^t, \theta^\star) - H(\theta^t, w^{t,k})\right) + \|w^{t,k+1} - w^{t,k}\|^2.$$

Hence, from Proposition 6, we obtain

$$\|w^{t,k+1} - \theta^\star\|^2 \le (1 - \alpha F_w) \|w^{t,k} - \theta^\star\|^2 + 2\alpha \left( \frac{F_\theta^2}{2F_w} \|\theta^t - \theta^\star\|^2 - \left(\frac{1}{\alpha} - \frac{L}{2}\right) \|w^{t,k+1} - w^{t,k}\|^2\right)$$
$$+ \|w^{t,k+1} - w^{t,k}\|^2$$
$$= (1 - \alpha F_w) \|w^{t,k} - \theta^\star\|^2 + \frac{\alpha F_\theta^2}{F_w} \|\theta^t - \theta^\star\|^2 - (2 - \alpha L) \|w^{t,k+1} - w^{t,k}\|^2,$$

which completes the first desired result.

Moreover, by specifically choosing the step-size $\alpha = 1/L$ we obtain that:

$$\|w^{t,k+1} - \theta^\star\|^2 \le \left(1 - \frac{F_w}{L}\right) \|w^{t,k} - \theta^\star\|^2 + \frac{F_\theta^2}{LF_w} \|\theta^t - \theta^\star\|^2 - \|w^{t,k+1} - w^{t,k}\|^2$$
$$\le \left(1 - \frac{F_w}{L}\right) \|w^{t,k} - \theta^\star\|^2 + \frac{F_\theta^2}{LF_w} \|\theta^t - \theta^\star\|^2.$$

This concludes the proof of this proposition. $\qquad \square$

Using these two results, we are ready to present the proof of Proposition 5

**Proposition 5.** Let $\{\theta^t\}_{t\in\mathbb{N}}$ be a sequence generated by Algorithm 2 with the step-size $\alpha = 1/L$. Then, we have:

$$\|\theta^{t+1} - \theta^\star\| \leq \sigma_K \|\theta^t - \theta^\star\| ,$$

where:

$$\sigma_K^2 \equiv (1-\kappa)^K \left(1-\eta^2\right) + \eta^2 .$$

with $\kappa \equiv L^{-1} F_w$ and $\eta \equiv F_w^{-1} F_\theta$.

*Proof.* Let $t \in \mathbb{N}$. From Proposition 7 (recall that $\alpha = 1/L$) and the fact that $\theta^{t+1} = w^{t,K}$, we have

$$
\begin{aligned}
\|\theta^{t+1} - \theta^\star\|^2 &= \|w^{t,K} - \theta^\star\|^2 \\
&\leq (1-\kappa) \|w^{t,K-1} - \theta^\star\|^2 + \eta^2\kappa\|\theta^t - \theta^\star\|^2 \\
&\leq (1-\kappa) \left[(1-\kappa)\|w^{t,K-2} - \theta^\star\|^2 + \eta^2\kappa\|\theta^t - \theta^\star\|^2\right] + \eta^2\kappa\|\theta^t - \theta^\star\|^2 \\
&= (1-\kappa)^2 \|w^{t,K-2} - \theta^\star\|^2 + \eta^2\kappa\left(1 + (1-\kappa)\right)\|\theta^t - \theta^\star\|^2 \\
&\leq \dots \\
&\leq (1-\kappa)^K \|w^{t,0} - \theta^\star\|^2 + \eta^2\kappa \sum_{k=0}^{K-1}(1-\kappa)^k \|\theta^t - \theta^\star\|^2 \\
&= (1-\kappa)^K \|\theta^t - \theta^\star\|^2 + \eta^2\kappa \sum_{k=0}^{K-1}(1-\kappa)^k \|\theta^t - \theta^\star\|^2,
\end{aligned}
$$

where the last inequality follows from the fact that $w^{t,0} = \theta^t$. Because $\kappa \in [0,1]$, the geometric series on the right hand side is convergent, and so we can write:

$$(1-\kappa)^K + \eta^2\kappa \sum_{k=0}^{K-1}(1-\kappa)^k = (1-\kappa)^K + \eta^2\kappa\frac{1 - (1-\kappa)^K}{1 - (1-\kappa)} = (1-\kappa)^K + \eta^2\left(1 - (1-\kappa)^K\right),$$

which completes the desired result. $\qquad\square$

Now, following our discussion in Section 6.3 about examples of loss functions that satisfy our assumption, we recall that here we focus on the following loss

$$H(\theta, w) = \frac{1}{2} \sum_s d(s) \sum_a \pi(a|s) \left( \mathbb{E}_{s'} \left[ r + \gamma \max_{a'} q(s', a', \theta) \right] - q(s, a, w) \right)^2.$$

As mentioned in Section 6.3 all is left is to show that Lipschitzness of $\nabla_w H(\theta, w)$ with respect to $\theta$.

**Proposition 8.** Assume that for any $(s, a)$, the function $\theta \to q(s, a, \theta)$ is $L_q(s, a)$-Lipschitz. Then, there exists $F_\theta > 0$ such that

$$\forall \theta_1, \forall \theta_2 \qquad \|\nabla_w H(\theta_1, w) - \nabla_w H(\theta_2, w)\| \leq F_\theta \|\theta_1 - \theta_2\|.$$

*Proof.* First, we compute the gradient $\nabla_w H$:

$$\nabla_w H(\theta, w) = \sum_s d(s) \sum_a \pi(a|s) \left( q(s, a, w) - \mathbb{E}_{s'} \left[ r + \gamma \max_{a'} q(s', a', \theta) \right] \right) \nabla q(s, a, w).$$

For the simplicity of the proof, we denote $Q(\theta_1, \theta_2) = \max_{a'} q(s', a', \theta_2) - \max_{a'} q(s', a', \theta_1)$. Hence

$$\|\nabla_w H(\theta_1, w) - \nabla_w H(\theta_2, w)\| = \gamma \left\| \sum_s d(s) \sum_a \pi(a|s) \left( \mathbb{E}_{s'} \left[ Q(\theta_1, \theta_2) \right] \right) \nabla q(s, a, w) \right\|$$

$$\leq \gamma \sum_s d(s) \sum_a \pi(a|s) \left| \mathbb{E}_{s'} \left[ Q(\theta_1, \theta_2) \right] \right| \|\nabla q(s, a, w)\|,$$

where we first cancelled the common parts in both gradient terms and the inequality follows immediately from the Cauchy-Schwartz inequality. Now, by using Jensen's inequality on the expectation, we obtain that

$$\|\nabla_w H(\theta_1, w) - \nabla_w H(\theta_2, w)\| \leq \gamma \sum_s d(s) \sum_a \pi(a|s) \mathbb{E}_{s'} \left[ |Q(\theta_1, \theta_2)| \right] \|\nabla q(s, a, w)\|.$$

Moreover, without the loss of generality we can assume that $\max_{a'} q(s', a', \theta_2) \geq \max_{a'} q(s', a', \theta_1)$, which means that we can remove the absolute value and obtain the following

$$\|\nabla_w H(\theta_1, w) - \nabla_w H(\theta_2, w)\| \leq \gamma \sum_s d(s) \sum_a \pi(a|s) \mathbb{E}_{s'} \left[ Q(\theta_1, \theta_2) \right] \|\nabla q(s, a, w)\|.$$

Now, by using a simple property of the maximum operation, we obviously have that

$$\max_{a'} q(s', a', \theta_2) - \max_{a'} q(s', a', \theta_1) \leq \max_{a'} \left( q(s', a', \theta_2) - q(s', a', \theta_1) \right).$$

Since the function $\theta \to q(s', a', \theta)$ is Lipschitz, we have that

$$q(s', a', \theta_2) - q(s', a', \theta_1) \leq L_q(s', a') \|\theta_1 - \theta_2\|.$$

It is also well-known that for such functions the gradient is bounded and therefore $\|\nabla q(s, a, w)\| \leq L_q(s, a)$. Combining all these facts yield

$$\|\nabla_w H(\theta_1, w) - \nabla_w H(\theta_2, w)\| \leq \gamma \sum_s d(s) \sum_a \pi(a|s) \mathbb{E}_{s'} \left[ Q(\theta_1, \theta_2) \right] \|\nabla q(s, a, w)\|$$

$$\leq \gamma \sum_s d(s) \sum_a \pi(a|s) \mathbb{E}_{s'} \left[ \max_{a'} \left( q(s', a', \theta_2) - q(s', a', \theta_1) \right) \right] \|\nabla q(s, a, w)\|$$

$$\leq \gamma \sum_s d(s) \sum_a \pi(a|s) \mathbb{E}_{s'} \left[ \max_{a'} L_q(s', a') \|\theta_1 - \theta_2\| \right] L_q(s, a)$$

$$\leq \gamma \|\theta_1 - \theta_2\| \mathbb{E}_{s'} \left[ \max_{a'} L_q(s', a')^2 \right] \sum_s d(s) \sum_a \pi(a|s)$$

$$= \gamma \|\theta_1 - \theta_2\| \mathbb{E}_{s'} \left[ \max_{a'} L_q(s', a')^2 \right],$$

where the last equality follows from the basic properties of the distribution matrix $D$. This proves the desired result. $\qquad \square$

