# OpenReview forum: "TD Convergence: An Optimization Perspective"
_NeurIPS.cc/2023/Conference — NeurIPS 2023 poster_

### Official Review · Reviewer_5ZUZ · 2023-06-18

**Soundness:** 4 excellent
**Presentation:** 3 good
**Contribution:** 2 fair
**Rating:** 6
**Confidence:** 4

**Summary:**

This work studies the TD learning algorithm from an optimization point of view which differs from the more classical fixed point Bellman operator point of view. The goal of the paper is to argue that this other viewpoint permits a better understanding of TD learning and a generalization of its convergence results explaining its practical success. The investigation of the known counterexample of TD divergence allows to identify the interplay of two forces that determine the  convergence behavior of TD learning, namely a so called target force and an optimization force. It is shown that TD shows a convergent behavior when the optimization force dominates the target one. These insights are then instantiated for linear function approximation with square loss and beyond under strong convexity and smoothness assumptions, even under the celebrated deadly triad.


**Strengths:**

- Understanding the behavior of the celebrated TD learning algorithm in the deadly triad setting and beyond the linear function approximation setting  is an important research goal given the popularity of the algorithm and its potential impact in RL.

- Interesting insights starting from the simple counter example in Section 4 are provided and clearly explained.

- The paper is very well-written, well-organized and overall easy to follow. To the best of my knowledge, proofs (including the appendix) are correct and cleanly presented.


**Weaknesses:**

1. It is not made very clear that the scheme proposed is actually different from the classical TD learning which was analyzed in [3] since the iterate $\theta_t$ of the target network is frozen. The paper considers a ‘target-based’ version of TD learning which was inspired by the DQN algorithm using target networks [1].

2. Since one of the motivations of the paper is to show that the optimization point of view allows to address more general settings than the linear function approximation setting with square loss, I would expect a more detailed discussion of these cases in Section 6 giving the definitions of the $H$ function in that case and verifying the uniform assumptions 1 and 2 of Section 6. The discussion in l. 277 to 280 is quite minimal. The generalization does still seem a bit restrictive and the analysis provides sufficient conditions that do not close the question of the understanding of the divergence behavior of TD learning. See also the Questions section.

3. Although the interpretation in terms of ‘target force’ and ‘optimization force’ has not been described as such in prior work to the best of my knowledge, arguments to show the results are quite classical and technical novelty is very limited in my opinion. The proofs of section 5 rely on the classical stability criteria in control for linear systems and was also used in other works ([16], see also discussions below about related works) even if the possibility of using other distributions instead of the stationary one was not described.The proofs for Section 6 follow standard analysis of gradient descent like algorithms for smooth and strongly convex objectives up to the drifting parameter $\theta_t$ which is periodically synchronized with the online iterate.

4. **Related work**: Closely related works are not discussed in detail, especially those analyzing the ‘target-based’ version of TD learning which is central in this work.

**(a)** While the present work indeed provides some new insights, the optimization perspective proposed in this work is not completely new and I believe some additional discussion regarding this would be welcome. As a matter of fact [16] is only briefly mentioned in l. 275 whereas the optimization point of view is alluded to in the remark in Section 2.4  of [16] (see also sections 3 and 5 therein) where the modified version of the Mean Square Bellman Operator with two variables  (target and online) clearly appear. While the generalization beyond linear function approximation and the squared loss (under some Lipschitzness and strong-convexity assumptions) and the flexibility to consider a different distribution from the Markov chain’s stationary distribution are interesting  insights, the results of Proposition 1 and Corollary 2 are not very novel. For instance, instead of periodically synchronizing the target network with the online one as in Algorithm 2, one could also consider the online moving average update rule proposed in the popular DDPG algorithm (Lillicrap et al. 2016) which was analyzed in the linear function approximation on-policy setting in [16, sections 2, 3, 5] and in Barakat et al. 2022 (see Sections 4.2, 5.1, 6.1).  The aforementioned results also provide almost convergence results and sample complexity analysis accounting for noisy settings unlike the present work.

Lillicrap, T. P., Hunt, J. J., Pritzel, A., Heess, N., Erez, T., Tassa, Y., Silver, D., and Wierstra, D. Continuous control with deep reinforcement learning, ICLR 2016.

Barakat, A., Bianchi, P., Lehmann, J. Analysis of a Target-Based Actor-Critic Algorithm with Linear Function Approximation, AISTATS 2022.

**(b)** If one of the main motivations or consequences of this work is to show that the ‘optimization viewpoint’ can explain the possible convergent nature of TD learning in the presence of the deadly triad as mentioned in the conclusion (under some assumptions and some suitable choice of the sampling distribution in the off-policy case), then, the ability of the target-based updates (which is actually the ‘optimization point of view’) was also advocated for in Zhang et al. 21 [25] at least in the linear function approximation setting (see Section 4 for off-policy evaluation with Q functions which could be easily adapted to V functions).

**(c)** Further works such as Liu and Olshevsky 2021 could also be relevant to mention. This work points out that original TD learning (i.e. Eq. (2), as proposed in [6] and analyzed in [3]) can be seen as what they call a ‘gradient splitting’ (see Section 3 therein) even if it is known that the TD learning update rule does not correspond to any gradient descent over any function.

Liu, R., Olshevsky, A. Temporal Difference Learning as Gradient Splitting, ICML 2021.

**(d)** Several works have also considered the analysis of TD learning with nonlinear function approximation beyond the linear setting (see e.g., Brandfonbrener and Bruna 2020; Agazzi and Lu 2021 to name a few). A discussion about these works seems also relevant given the generalization motivation of the present work.

Brandfonbrener, D., Bruna, J. Geometric insights into the convergence of nonlinear TD learning, ICLR 2020.

Agazzi, A., Lu, J. Temporal-difference learning with nonlinear function approximation: lazy training and mean field regimes, MSML 2021.

**Minor typos:**
l. 104: ‘the root cause of TD’, of divergence?
l. 242: capital $H$ instead of $h$
l. 541-542: $\theta^{\star}$ instead of  $\theta^{*}$


**Questions:**

In the following, I list a few questions of which the first two are the main ones, focusing on the limit point definition and the strength of the assumptions.

1. All the theoretical results show convergence to the fixed-point $\theta^{\star}$. How is this point defined in those results?  Eq. (5) mentions that if convergence happens then $\nabla_{w} H(\theta^{\star}, \theta^{\star}) = 0$.  Such a characterization (which is also the fixed-point characterization of the TD solution) is used in all the proofs. Then l. 142-143 precise that ‘Whenever it exists, we refer to $\theta^{\star}$ as the fixed point of these iterative algorithms’. For the convergence results to be meaningful, the existence of the fixed point should be guaranteed. I guess you define $\theta^{\star}$ to be the fixed point of the projected Bellman operator which is indeed a contraction under some conditions but this is not very clear in the paper. However, as alluded to in the paper in l. 212-218, it is not clear whether the projected Bellman operator is still a contraction when one considers a distribution different from the stationary state distribution of the Markov Chain. What would then be the limit point(s) in the results in the case where existence is not guaranteed by the fixed point arguments relying on the operator viewpoint? Do you just suppose in that case that there exists a unique point such that $\nabla_{w} H(\theta^{\star}, \theta^{\star}) = 0$ (which is what is required and used to conduct the proofs)?

2. Concerning the assumptions and the examples provided in Section 6, input-convex neural networks [18] provide convex functions with respect to the inputs and not with respect to their weights (parameters). How these would help to guarantee that the strong convexity assumption 2 holds? Satisfying both assumptions does not seem straightforward. The constants $F_{\theta}$ and $F_{\omega}$ are uniform constants over $\theta$ and $w$ that are not easy to define and compute in practice which makes the core stability condition $F_{\theta} < F_{w}$ difficult to verify. I understand though that the focus of the paper is theoretical. Even in the linear setting and assuming that the feature matrix is given and known, can we for example suggest some distributions beyond the stationary state distribution for which the condition holds?

3. Regarding section 4, when you mention that the counter example was identified by [3], are you referring to Section IX in Tsitsiklis and Van Roy 97 (TAC) for this? Is it a simplified/modified example of that one?

4. Could you mention the stationary state distribution of the Markov chain for the counter example? This could be added if relevant to show that this stationary distribution is indeed valid and leads to a convergent behavior as expected.


**Limitations:**

Beyond the points raised above, one limitation that is not clearly mentioned is that the convergence results are limited to the deterministic setting.

---

> ### Author Rebuttal · Authors · 2023-08-09
>
> We thank the reviewer for recognizing that we have tackled an important research goal, for stating that our paper is insightful, clearly-written and well-organized, and also for your diligence in checking our proofs.
>
> In terms of weakness 1, we will emphasize better that one of the major contributions of our paper is to show convergence of TD under a frozen target network ($K>1$). In fact, our results include the special case (with $K=1$), but also bridges the gap with practice where usually a much larger value of $K$ is used. We will put more effort to emphasize this.
>
> In terms of discussing examples that satisfy our assumptions, Please see our detailed discussion in the general rebuttal. We will add this to the paper.
>
> In terms of weakness 3, the novelty of our results, we respectfully disagree that the proof techniques are standard. In fact, by following the standard proof techniques of gradient descent (which Lee and He 2019 leveraged) they end up with an error term that accumulates over iterations, and so exact convergence to TD fixed-point cannot be shown with a finite $K$. Using proof techniques that are new to the RL literature, we showed for the first time that irregardless of the value of $K$, and under the natural extension of quadratic loss, contraction can be shown, and convergence to exactly the TD fixed-point can be concluded. To the best of our knowledge, this is a quite novel and significant.
>
> As for Section 5 about linear approximatiors and squared loss, our intention was not to say that we are very novel in exploring this setting (we provided credit to van Roy Tsitsiklis and other influential papers). Our intention was merely to formalize our intuitions from the counter example, before moving to the general setting in Section 6 and present our main novel result. We are happy to clarify this in the paper.
>
> In terms of related work, we agree that more discussion will strengthen the paper. We extensively discussed relationship to two of your suggested papers in our general rebuttal and in response to reviewer 16Sb. Here, we discuss a third paper you mentioned and defer further discussion to the paper due to space limits. In the case of gradient splitting work of Liu and Olshevsky. Their contribution is to introduce the notion that the update of the TD algorithm with $K=1$ could be thought as gradient splitting. Leveraging this insight, and by adding a projection step to TD, they improved the sample complexity bound of Bhandari et al (which we have cited in the submission). The key difference with our case is that we are interested in vanilla TD convergence under general $K$ and also that we have extended the quadratic loss to the somewhat more general case of strongly-convex functions. Nevertheless, we found this work quite interesting and will add this to the paper. Thanks.
>
> Please rest assured that based on our investigation the other papers you mentioned, while quite innovative, do not undermine the novelty of our work. We will cite and discuss them, and we hope the reviewer considers increasing their score in light of our better situating our paper.
>
> Regarding the existence of a fixed point, we want to make a distinction between existence of a fixed-point and the fact that a certain operator is a contraction. For example, in the counter example a fixed-point always exists (namely $\theta^{\star}=0$), for which $\nabla_{w} H(\theta^{\star},\theta^{\star})={\bf 0}$. This is despite the fact that for some distributions TD might be divergent. So, the existence of a fixed-point is a property of the problem, not the algorithm that attempts to find the solution to that problem.
>
> More generally, in the linear case with quadratic loss we can show that a fixed-point always exists under very mild assumptions. In particular, from $\nabla_{w} H(\theta^{\star},\theta^{\star})=0$ we get:
>
> $\Phi^{\top} D(R+\gamma P\Phi\theta^{\star} - \Phi\theta^{\star})=0,$
>
> meaning:
>
> $\theta^{\star} = \big(\Phi^{\top}(I - \gamma P)\Phi\big)^{-1}\Phi^{\top} D R.$
>
> $P$ is a stochastic matrix, so $(I-\gamma P)\succ 0$, therefore the inverse always exists and the problem always has a unique fixed-point. Some algorithms might diverge in their attempt to find the fixed point, but that does not change the fact that it exists.
>
> More generally it is an open question whether there exists a fixed-point with alternative $H$. We generally need additional information to answer this question. In the optimization literature determining if a problem has a solution is a vibrant research area. However, this is beyond the scope of this paper, and we would love to explore this direction once this paper gets accepted.
>
> To clarify our pointer to input-convex neural nets, the reviewer is correct in saying that the original paper focused on the convexity of the net with respect to the input given fixed weights. That said, the same network architecture the paper proposed is also convex with respect to the weights given a fixed input. To see this, consider the class of functions $f(x,\theta)=ReLU(x^{\top}\theta)$. Notice that this function is both a) convex with respect to $x$ given fixed $\theta$, and b) convex with respect to $\theta$ given fixed $x$. Thank you for your astute question, and we will clarify this in the paper.
>
> Regarding the counter example, we took it from the standard textbook of Sutton and Barto. In their Example 11.1, Sutton and Barto credit it to van Roy and Tsitsiklis and we followed the standard they set.
>
> Regarding the stationary-state of the example, the distribution depends on the parameter $\varepsilon$ used to define the transition model. In particular, it assigns $\frac{\varepsilon}{1+\varepsilon}$ to the first state, and $\frac{1}{1+\varepsilon}$ to the second state. We add this to the paper.
>
> You are right about the deterministic limitation. To clarify, we support stochastic transition matrices, but that the gradient computation needs to be exact (deterministic). We will add this limitation.

---

> > ### Comment · Reviewer_5ZUZ · 2023-08-15
> > **post rebuttal**
> >
> >
> > I thank the authors for their detailed responses which answered all my questions, especially for additional comments regarding examples when the Lipschitzness and strong convexity assumptions are satisfied as well as the contextualization of the work within the recent literature. I believe some clarifications provided by the rebuttal are worth adding to the paper. Although I still think the setting is a bit restrictive as a generalization beyond the standard linear function approximation setting (in the deterministic case), I raise my score to 6 given the authors’ rebuttal and the comments regarding off-policy/on-policy convergence.
> >
> > Regarding the existence of a fixed point, the sense of my comment was about conditions to ensure that such a point that is central to the results and their proofs exists in the more general setting which concerns this work. It is known that a fixed point exists in the linear case with quadratic loss (namely the standard TD solution in the linear function approximation setting) and that this point can be reached since the Bellman contraction allows to design even stochastic approximation algorithms (namely TD learning) to reach this point. I would have expected to see at least an example beyond the known more standard case where this fixed point is guaranteed to exist. My current understanding is that you suppose that there exists such point throughout the paper, I understand though that this may be treated on a case by case basis depending on the specific $H$.
> >
> > On a more technical point concerning the input-convex neural nets, the ReLu example provided in the response introduces some nonsmoothness which does not seem to be strictly covered by the assumptions (maybe some smoothing preserving convexity could address this technical point though).

---

> > > ### Author Response · Authors · 2023-08-16
> > > **Thanks**
> > >
> > > We are delighted to hear that the reviewer found the discussions helpful, and we will be sure to add these discussions to the paper. We also do believe that the discussion phase truly made our paper much stronger. While the strong convexity assumption is a natural extension of the existing literature, we also do agree that it is not the most general setting. We hope that the publication of this paper could instigate further research in this direction to further generalize TD convergence using the optimization point of view.
> > >
> > > Note that in providing the ReLU example our intention was to primarily argue that an input-convex neural network is also convex with respect to the weights. The reviewer is correct in stating that the ReLU activation, even though it is Lipschitz, may still be problematic because the gradient is undefined at 0. In this case we need to resort to smoothing, as you mentioned, or use alternative activation functions such as softplus.
> > >
> > > We highly appreciate the raised score!

---

### Official Review · Reviewer_fasm · 2023-07-04

**Soundness:** 4 excellent
**Presentation:** 4 excellent
**Contribution:** 3 good
**Rating:** 7
**Confidence:** 4

**Summary:**

This paper studies convergence of the TD algorithm from the perspective of solving a shifting optimization problem. Through a classic failure case, the authors uncover two forces, whose interplay reveals TD's convergence properties. These two forces both depend on the state visitation distribution, the state features, and the transition kernel. The authors point out that while the stationary-state distribution ensures convergence, this does not mean no other state visitation distributions can and seek to establish sufficient properties. They generalize their analysis to TD error defined with more general functions and provide these sufficient conditions. Assuming the TD error $H$ has a value function gradient that is Lipschitz in the target function parameters and is also strongly convex in the value function parameters, a more general convergence criterion can be derived. The authors extend this result to the setting where the shifting optimization problem is only solved approximately at each step using $K$ gradient updates.

**Strengths:**

I found this paper to be very clearly written. To my knowledge, the claim in Proposition 1 and subsequent results are novel and make significant progress towards understanding convergence of the TD algorithm.

**Weaknesses:**

The paper does a great job explaining and motivating the novel TD convergence analysis, and I believe it can stand on its own as a "theory paper". But I think experimental support for the approach in a complex (e.g., deep RL) setting could have made the paper much stronger.

I am also not entirely clear which results are novel and can be attributed to the authors vs which were known beforehand. Equation (7) does not appear to be novel; some form exists in [35], sec 4. The paper could benefit from better signposting to explain where others got stuck and this paper advances.

**Questions:**

- line 164: where does the $1/2$ come from in the equation? If that is there for convenience, can you add a $1/2$ to the def of $H$ under line 117?
- line 237: $M_w$ is positive definite if $\Phi$ is full rank **AND** $D$ is full-rank, i.e., every state has positive probability under $d(s)$, no?
- line 320: $L$ here is typically known as the strong-smoothness parameter, no? In contrast to the strong-convexity parameter.
- line 330: the condition number I'm familiar with (see [1]) is always greater than or equal to $1$ (max eig / min eig). In this case, I think you mean the "inverse condition number". Also, this technically means, $\sigma_K^2 = 1$ when $\kappa=1$, in which case, the analysis does not imply convergence. Can you please discuss this corner case?
- line 370: "exits" --> "exists"

[1] Guille-Escuret, Charles, et al. "A study of condition numbers for first-order optimization." International Conference on Artificial Intelligence and Statistics. PMLR, 2021.
http://proceedings.mlr.press/v130/guille-escuret21a/guille-escuret21a.pdf

**Limitations:**

I see no need for discussion of negative societal impact in this work.

---

> ### Author Rebuttal · Authors · 2023-08-09
>
> We appreciate the reviewer's carefully reading our paper as well as the overall quite positive assessment. Thanks for pointing out that the paper is clearly written and also thanks for recognizing the novelty of our work in terms of extending TD convergence proof to general $K$, as well as extending the function class from quadratic to other alternatives.
>
> - The paper does a great job explaining and motivating the novel TD convergence analysis, and I believe it can stand on its own as a "theory paper". But I think experimental support for the approach in a complex (e.g., deep RL) setting could have made the paper much stronger.
>
> In deriving a generalized proof for TD, our motivation was to be able to explain the remarkable empirical success of TD for general $K$ and beyond the linear function approximators and in settings such as deep RL. We liked our theory to complement the existing evidence on the solid empirical performance of TD in the literature, and to provide further theoretical evidence that TD is a sound algorithm in a broader setting than understood in previous work.
>
> - I am also not entirely clear which results are novel and can be attributed to the authors vs which were known beforehand
>
> Thanks for highlighting this. We would like to summarize here the two main contributions of our paper, which we will better highlight in the main paper as well:
>
> 1- To the best of our knowledge, we are the first paper to show the contraction result of TD with general value of $K$. We show contraction with convergence to exactly the TD fixed point. In this space, closest to our work is that of Lee and He (2019), which could only show that TD with general $K$ converges to a region around the fixed-point. The part where they got stuck is where they use the more classical analysis that gradient descent only approximately solves each iteration, and so they need to accumulate some error along the way. In contrast, we can show that even approximately solving each iteration (corresponding to finite $K$) is enough to obtain contraction (without any error term). Even though Lee and He are correct in saying that using a finite $K$ results in approximately solving each iteration, we can still show that each iteration remains a contraction by looking at the net effect of the  updates to the online network and the single update to the target network. We prove that the net effect of these updates is one that ensures that the iterate makes steady progress towards the unique fixed-point irregardless of $K$.
>
> 2- To the best of our knowledge, we are also the first paper to show TD convergence in the most natural extension of quadratic functions, namely the strongly convex case. This allows us to argue that slight modifications of TD in terms of loss functions and function approximators are also sound so long as they satisfy our assumptions. We are unaware of any previous work that tackled this extension.
>
> - where does the $\frac{1}{2}$ come from in the equation?
>
> Yes, you are correct that this was added for convenience and we should also add it in line 117. Thanks for catching the missing $\frac{1}{2}$.
>
> - $M_{w}$ is positive definite if $\Phi$ is full rank AND $D$ is full-rank, i.e., every state has positive probability under, no?
>
> You are correct, and we will clarify this in the paper. Thanks for your diligence.
>
> - $L$ here is typically known as the strong-smoothness parameter, no? In contrast to the strong-convexity parameter.
>
> We think the reviewer means ``global'' Lipschitz continuous, so yes you are correct, and we will clarify this.
>
> - the condition number I'm familiar with (see [1]) is always greater than or equal to  (max eig / min eig). In this case, I think you mean the "inverse condition number". Also, this technically means, $\sigma_K^2=1$, when , $\kappa=1$ in which case, the analysis does not imply convergence. Can you please discuss this corner case?
>
> You are right in saying that we mean the inverse condition number here. In terms of the corner case, notice that if $\kappa=1$ (which means we have a quadratic dependence to the online parameter), then $\sigma_k=\eta$, which given the assumption is less than one. Therefore, this is a contraction and convergence will be guaranteed.
>
> - line 370: "exits" --> "exists"
>
> We will fix it. Thanks!

---

> > ### Comment · Reviewer_fasm · 2023-08-11
> > **Acknowledgement of Rebuttal**
> >
> > Dear authors, thank you for your rebuttal. You have addressed my concerns. I understand your point of supporting the performance of TD with general $K$ with citations rather than your own experiments (although reproducing others results never hurts). I also see that your analysis in the case where gradient descent is run for $K$ steps is a key contribution. I have read through the other reviews and see that you have emphasized this point in your rebuttals to them as well. I will continue to follow those dialogues, but for now, I maintain my score.

---

> > > ### Author Response · Authors · 2023-08-11
> > > **Thank you**
> > >
> > > We appreciate your carefully reading our paper, and your engagement with our rebuttal. We also thank you for the continued support.
> > >
> > > If there was any lingering or new question, please do not hesitate to bring it into our attention.

---

### Official Review · Reviewer_gJiK · 2023-07-06

**Soundness:** 3 good
**Presentation:** 3 good
**Contribution:** 1 poor
**Rating:** 7
**Confidence:** 4

**Summary:**

The paper studies TD-learning with target network update. The authors recast the TD-learning algorithm into a time-varying optimization problem. The authors proves convergence for a function class with strong convexity and smoothness.

**Strengths:**

The paper is easy to follow and the motivation of the work is well explained by a simple example form $\theta\to2\theta$. Moreover, the viewpoint of target force and optimization force seems to novel viewpoint, and the theoretical result seems to be solid.

**Weaknesses:**

1. Assuming strong convexity and lipschitzness is too restrictive to argue for a general function class. Moreover, regarding the condition $F_{\theta}<F_{w}$ in Theorem 3, I believe this is the key condition for the convergence but the discussion seems to be missing whether it is a common condition to be met or not.

2. The analysis on iterative optimization objective for tabular and linear case has been studied in Lee et al. and Zhang et al.. The comparison with the existing work seems to be insufficient.

**Questions:**

1. How strict is the condition $F_{\theta}<F_{w}$ in Theorem 3? Can we find any examples other than tabular or linear setting to show the convergence under general setting?

2. Can we also find conditions for to ensure the convergence for the Baird example?

**Limitations:**

In summary, the impact of theoretical result is not sufficient for the following reasons:

- Assumption on strong convexity is too strong.

- There are no examples or experiments showing convergence of general function class other than tabular or linear setting.

- There are no compariosn between the existing works Lee et al., and Chen et al..

Hence, I am leaning towards rejection as for now.

Lee, Donghwan, and Niao He. "Target-based temporal-difference learning." International Conference on Machine Learning. PMLR, 2019.

Chen, Zaiwei, John Paul Clarke, and Siva Theja Maguluri. "Target Network and Truncation Overcome The Deadly Triad in $ Q $-Learning." arXiv preprint arXiv:2203.02628 (2022).

---

> ### Author Rebuttal · Authors · 2023-08-09
>
> We thank the reviewer for the review. In what follows, we address the particular weaknesses and questions raised.
>
> - A discussion seems to be missing whether it is a common condition to be met or not.
>
> Please see our detailed discussion in the general comment part.
>
> - The analysis on iterative optimization objective for tabular and linear case has been studied in Lee et al. and Zhang et al.. The comparison with the existing work seems to be insufficient.
>
> We answered this question in depth in our general comment, but we are happy to distill our point here so we better situate our result in comparison to these two papers mentioned by the reviewer. Starting by Lee and He (2019), note that while we were able to show TD convergence, Lee and He can only guarantee that TD will find a solution in a region around the fixed-point because with finite $K$ the analysis needs to account for errors that are accumulated in each iteration. Another way to think about their result is that they can only show contraction if one uses gradient descent with infinite $K$. With finite $K$, they need to account for errors in solving each iteration (this is denoted by $\epsilon_k$ in their proofs such as in Theorem 3). To better make the point, suppose that we solve the first iteration with some error, meaning that we approximately solve:
>
> $\theta^{1} \approx \arg\min_{w} H(\theta^0,w),$
>
> but then after we solve each iteration perfectly, meaning for $i\geq 1$:
>
> $\theta^{i+1} =  \arg\min_{w} H(\theta^i,w).$
>
> Clearly, one can think of approximately solving the first iteration as initializing TD to a point different than $\theta^0$ and then doing perfect optimization in all iterations. The fact that we then can solve each subsequent iteration perfectly should give us exactly a contraction to the TD fixed-point based on our analysis. However, in this case the result of Lee and He (2019) can still only support convergence to a neighborhood characterized by the approximation error $\varepsilon$. In contrast, in this case we can guarantee convergence exactly to the TD fixed-point.
>
> Moreover, in light of the more practical literature on TD where a finite $K$ is used, we were interested to show that 1) we converge to the TD fixed-point exactly and 2) with any value of $K$ the TD algorithm would give us a contraction. We indeed show in our main result that TD is a contraction with any value of $K$ (which we believe we are the first paper to show). More concretely, we show that smaller values of $K$ can damage the contraction factor, but each iteration will nevertheless remain contractive irregardless of $K$.
>
> In terms of comparison with Zhang et al. (2021) and Chen et al. (2022), notice that due to the difficulties pertaining to proving convergence for vanilla TD, a line of existing research was to equip TD with modifications to make it more conducive to convergence. In this case, Zhang et al. (2021) introduced two projection steps that are crucial for obtaining convergence, and similarly Chen et al. (2022) studies the case where a truncation step is added. These are very important techniques, and they indeed can make TD more convergent. However, our convergence result does not lean on these projection and truncation steps and is applicable to vanilla TD as well as TD with alternative loss functions and function approximators.
>
> - Can we find any examples other than tabular or linear setting to show the convergence under general setting?
>
> Please see our detailed discussion in the general comment part for more examples.
>
> - Assumption on strong convexity is too strong.
>
> Notice that the existing literature, including all the three papers mentioned by the reviewer, revolve around the linear function approximation case with quadratic loss function. This is just a special case of having a strongly-convex function. So, to the best of our knowledge, we are showing TD convergence in a setting that is less stringent and more general than the previous work. We would like to highlight that one can only hope to understand the more difficult cases (such as convex, weakly convex, and non-convex) by first understanding the most natural extension of the quadratic loss, namely the strongly-convex case. This was absent in existing work and our goal was to fill this gap.
>
> - There are no examples or experiments showing convergence of general function class other than tabular or linear setting.
>
> Again, please see our detailed discussion in the general comment part for more examples.

---

> > ### Comment · Reviewer_gJiK · 2023-08-10
> >
> > Thank you for the detailed response. My main concerns regarding the comparison with existing works have been mostly addressed. However, I have still have some remaining concerns regarding the condition $F_{\theta}<F_w$. This condition seems to be quite related to the assumption, equation (7) in Thereom 1 in Melo et.al.. That is, the condition $F_{\theta}<F_w$ can be met when we have certain strong assumptions on behavior policy and target policy. Besides, the condition (7) in Thereom 1 in Melo et.al., has factor of $\gamma^2$ whereas the proof in the attached rebuttal requires $\gamma$ , which implies stricter condition.
> >
> > Melo, Francisco S., Sean P. Meyn, and M. Isabel Ribeiro. "An analysis of reinforcement learning with function approximation." Proceedings of the 25th international conference on Machine learning. 2008.

---

> > > ### Author Response · Authors · 2023-08-10
> > > **Situating Our Paper Relative to Melo et al.**
> > >
> > > We again thank the reviewer for carefully reading our paper. We highly appreciate the reviewer's engagement with the clarifications we made in our rebuttal.
> > >
> > > Thanks for bringing the work of Melo et al (2008) into our attention. While we did cite this paper in our original submission (reference [22]), we agree that a more nuanced discussion about this important paper is warranted. We distill the key differences here, and in particular we discuss the similarity between their condition (7) and our condition. This discussion will be added to the paper.
> > >
> > > First, we understand the key contribution of Melo et al (2008) to be the extension of the ODE proof of Tsitsiklis and Van Roy (1997) from TD prediction to the more general control setting with Q-learning. Melo et al. show that asymptotically and under mild assumptions, the Q-learning algorithm with linear function approximation, quadratic loss, and $K=1$ converges to a fixed-point.
> > >
> > > In this context, and to answer the reviewer's specific question, the condition in their equation (7) is in fact quite different than the condition identified in our paper. In their equation (7), they require that the eigen values of $\Sigma_{\pi}$, the policy-conditioned covariance matrix $\Phi^{\top} D\Phi$, dominate the eigen values of a second matrix $\gamma^{2} {\Sigma_{\pi}^{\star}}(\theta)$. Here $\Sigma^{\star}_\pi(\theta)$ is a similar covariance matrix for features, but one that is computed based on the action-greedification step.
> > >
> > > While this condition may look similar, a deeper investigation reveals that it is completely different from our condition. To recap, our condition requires that the optimization force (namely the same covariance matrix of the features) dominate the target force. In this case, the target force is defined as the degree to which the objective function $H$ can be affected by the target network variable. In the linear case, it depends on the eigen values of the matrix $\gamma \Phi^{\top} D P \Phi$. Notice that this target force is governed, in part, by the transition matrix $P$. This is a major distinction between our condition and that of Melo et al. whose condition does not depend on $P$, and intuitively, their condition is more related to choosing the feature matrix $\Phi$ in such a way as to contain the negative effects due to maximization. To conclude, because the conditions are inherently different, the $\gamma^2$ factor in their result cannot be thought of as offering any particular advantage relative to our result.
> > >
> > > Moving from the condition itself, a major difference is present in the final guarantees obtained by us relative to Melo et al. In our case, we are interested in finding a finite-time contraction result, meaning that after each outer iteration $t$, we would like to provide a guarantee on the quality of the iterate we find relative to the TD fixed-point, whereas Melo et al. showed the somewhat weaker asymptotic convergence.
> > >
> > > Moreover, our results are better grounded in the empirical side of the RL literature in two important ways, namely 1) our results support the now ubiquitous deep RL practice of freezing the target network (which corresponds to the case of $K>1$) whereas Melo et al. study the case of $K=1$, and 2) we generalized the function class beyond the case of just linear functions and quadratic loss, and support a more abstract function class $H(\theta,w)$. This abstract class includes the setting studied by the important work of Melo et al. as a special case, but our setting also includes other interesting examples as explained in our general rebuttal.
> > >
> > > We are delighted to see that our rebuttal already addressed your main concerns, and we hope you find the new discussion on Melo et al. useful as well. We really hope that you take the addressed concerns into account and raise your score to support our paper.

---

> > > > ### Comment · Reviewer_gJiK · 2023-08-11
> > > >
> > > > Thank you for the response. Indeed, as the authors mentioned, the condition with Melo et al. is different. However, my point was that $F_{\theta}<F_w$ somehow requires strong condition on the off-policy distribution. For example, as the authors mentioned, for linear case, $F_{\theta} =  \lambda_{\max}(\gamma \Phi^{\top}PD\Phi)$ and $F_w = \lambda_{\min}(\Phi^{\top}D\Phi)$, which is equivalent to the condition $\gamma \Phi^{\top}PD\Phi < \Phi^{\top}D\Phi $. This is exactly the condition corresponding to the on-policy case, which is also mentioned in the paper. Even though, this condition can be met for other than the stationary distribution case, the condition, $ \gamma \Phi^{\top}PD\Phi < \Phi^{\top}D\Phi $,  itself is usually important. Overall, my concerns have been mostly addressed, and I would like to increase my score.

---

> > > > > ### Author Response · Authors · 2023-08-11
> > > > > **Thank you**
> > > > >
> > > > > You are exactly right.
> > > > >
> > > > > In our view, one of the most important open questions is to identify, given a $P$ and $\Phi$, the kind of off-policy distributions $D$ that still result in satisfying the condition. For example with the counter example, we saw that with some off-policy distributions the contraction factor, which is just $\frac{F_\theta}{F_w}$ gets even smaller than the contraction factor with the on-policy distribution. That means convergence under such off-policy distributions is actually even faster than convergence under the on-policy distribution! Of course in general identifying such nice distributions is a non-trivial task, and so we really hope that this paper can instigate further research in this direction.
> > > > >
> > > > > We are, again, highly appreciative of the reviewer's engagement with our rebuttal and for their supporting our paper!

---

### Official Review · Reviewer_16Sb · 2023-07-23

**Soundness:** 2 fair
**Presentation:** 3 good
**Contribution:** 2 fair
**Rating:** 7
**Confidence:** 4

**Summary:**

The paper studies the convergence conditions for Temporal Difference (TD) learning utilizing a target network, and further extends its findings to scenarios where TD minimizes alternative losses beyond mean square errors. The study is conducted by formulating TD updates as iterative optimizations, under the help of a target network. Notably, the paper provides an intuitive description of the coefficients before the student's TD network parameter and the target network parameter as "optimization force" and "target force," respectively. The paper reaches the conclusion that when optimization force dominants, the algorithm converges.

**Strengths:**

(1) The paper is of high quality and clarity. The authors have provided a clear setting, accompanied by well-presented proofs and well-defined assumptions. The demonstration of the counterexample is worth mentioning, as it effectively and intuitively introduces the main idea of the paper.

(2) The paper extends current studies of TD to cases where alternative losses, such as Huber loss, are used, closing a gap between the theory and practical algorithms.

**Weaknesses:**

(1) The paper focuses on Markov Reward Process, which is not a common setting for TD convergence proof. Why authors did not focus on expected updates? Could authors provide more reasoning for their choice?

(2) The paper re-forms TD updates as iteration optimizations with the help of a target network. Could authors add more comparisons to the convergence results of TD with a target network? For example,
Breaking the Deadly Triad with a Target Network, Zhang et al. (2021)
Target-Based Temporal-Difference Learning, Lee and He (2019)

(3) Some empirical results to show that the empirical contraction factor aligns with their theory findings would be great.

(4) The paper proposes an interesting point: for some safe state distribution, TD converges in the off-policy case. Could authors provide the analytical form of safe distributions? Also, how should we compute these distributions in practice?

**Questions:**

(1) Do Huber loss, logistic loss and entropy loss satisfy all three assumption stated in Section 6 for inexact approximation?

**Limitations:**

The paper did not discuss the limitations. Some discussions on an extension to stochastic updates and non-linear settings would be fascinating.

---

> ### Author Rebuttal · Authors · 2023-08-09
>
> We thank the reviewer for the time spent carefully reviewing the paper and for appreciating our work. Please find below some clarification regarding your questions.
>
> - The paper focuses on Markov Reward Process, which is not a common setting for TD convergence proof. Why authors did not focus on expected updates? Could authors provide more reasoning for their choice?
>
> Thank you for highlighting this question. In our case, we just followed the MRP setting studied in Chapter 11 of Sutton and Barto (specifically Example 11.1). Having examined our results based on your question, we found that from the optimization point of view, studying TD in this MRP setting is identical to the setting where we look at the expected TD update. Therefore, our results can be framed in the expected setting and we are happy to add this framing to the paper.
>
> - Could authors add more comparisons to the convergence results of TD with a target network? For example, Breaking the Deadly Triad with a Target Network, Zhang et al. (2021) Target-Based Temporal-Difference Learning, Lee and He (2019)
>
> Sure. We start by better situating our paper with respect to Lee and He (2019). While previous work primarily focused on $K=1$ (the setting of changing the target network after each update to the online network) the work of Lee and He (2019) was, to the best of our knowledge, the first paper that tackled general $K$. One limitation of their result is that they could only show that TD will find a solution in a region around the fixed-point because with finite $K$ the analysis needs to account for errors that are accumulated in each iteration. Another way to think about their result is that they can only show contraction if one uses gradient descent with infinite $K$ (which corresponds to exactly solving each iteration). With finite $K$, they need to account for errors in solving each iteration (denoted by $\epsilon_k$ in their proofs such as in Theorem 3), which prevented them from obtaining an exact convergence result.
>
> In contrast, in light of the more practical literature on TD where a finite $K$ is used, we were interested to show that 1) we converge to the TD fixed-point exactly and 2) with any finite value of $K$, the TD algorithm would give us a contraction. We indeed show in our main result that TD is a contraction with any value of $K$, which we believe we are the first paper to show this kind of result in the RL literature. More concretely, we show that smaller values of $K$ can damage the contraction factor, but each iteration will nevertheless remain contractive irregardless of $K$.
>
> In terms of comparison with Zhang et al. (2021), notice that due to the difficulties pertaining to proving convergence for vanilla TD, a line of existing research was to equip TD with modifications to make it more conducive to convergence. In this case, Zhang et al. (2021) introduced two projection steps that are crucial for obtaining convergence. However, we were able to obtain a general result showing that vanilla TD converges with any value of $K$, and it also supports TD convergence in a broader setting.
>
> - Some empirical results to show that the empirical contraction factor aligns with their theory findings would be great.
>
> In deriving a generalized proof for TD, our motivation was to be able to explain the remarkable empirical success of TD with general $K$ and beyond linear function approximators. We liked our theory to complement the existing evidence on the solid empirical performance of TD in the literature, and to provide further theoretical evidence that TD is a sound algorithm in a broader setting than understood in previous work.
>
> - The paper proposes an interesting point: for some safe state distribution, TD converges in the off-policy case. Could authors provide the analytical form of safe distributions? Also, how should we compute these distributions in practice?
>
> This is in fact a very important open question. We note that the fact that we are now interested in this open question is an insight that is driven from our optimization perspective. We believe that highlighting this open question, as noticed by the reviewer, is in fact a merit and not a weakness of our work. We believe that this question should be investigated deeply in future work, and that while quite important, remains outside of the scope of our paper.
>
> - Do Huber loss, logistic loss and entropy loss satisfy all three assumptions stated in Section 6 for inexact approximation?
>
> Please see our detailed discussion in the general comment part.
>
> - Some discussions on an extension to stochastic updates and non-linear settings would be fascinating
>
> Again, please see our detailed discussion in the general comment part about non-linear examples. Also, based on our preliminary investigation our theory can be extended to the case with stochastic gradients, and so publishing this paper will open the gate for a more thorough investigation of this case and beyond.

---

> ### Comment · Reviewer_16Sb · 2023-08-12
> **Thanks for the detailed explanations!**
>
> Most of my questions are answered, especially my concerns on comparison to other target-based papers and showing examples satisfying the assumptions. Meanwhile, the discussion with Reviewer gJiK on when $F_{\theta} < F_{\omega}$ is inspiring. It would be great if the condition can be presented more straightforwardly, for example, directly stating the conditions on the feature matrix and state distribution.
>
> I have raised my score from 5 to 7.

---

> > ### Author Response · Authors · 2023-08-15
> > **Thanks**
> >
> > We are delighted to see that the reviewer found it helpful to read our discussion on 1- comparison to other target-based papers 2- satisfying assumptions and 3- off-policy distributions that might achieve better contraction factors. We will add these discussion to the paper, and will also present them in a more straightforward manner.
> >
> > Thanks for your continued support, and for raising your score. We highly appreciate it!

---

### Author Rebuttal · Authors · 2023-08-09

We appreciate the thoughtful feedback provided to us by our reviewers. All reviewers agreed that our results are clearly articulated. Notably, Reviewer 16Sb believes that the paper is of high quality and is accompanied by well-presented proofs. Also, Reviewer fasm confirms that our results are novel and make significant progress towards a better understanding of TD convergence.

In light of the reviews, we realized that we could have done a better job in articulating the major contributions of our work. We reiterate here that our paper has made two significant advancements in terms of generalizing existing results on TD convergence:

1- We believe to be the first paper to show a contraction for TD with frozen target network and general $K$. To elaborate further, to the best of our knowledge, existing results prior to Lee and He (2019) mainly considered the case where we either never freeze the target network (corresponding to the value of $K=1$), or the somewhat unrealistic case where we can exactly solve each iteration. Lee and He (2019) showed guarantees pertaining to the more general case of finite $K>1$, but notice that, while their result is quite innovative, they leaned on the more standard optimization tools for ensuring that gradient descent with a fixed $K$ can only solve each iteration approximately. Therefore, each iteration results in some error. In their theory this error is accumulated per iteration and needs to be accounted for in the final result. Therefore, they fell short of showing 1) contraction and 2) exact convergence to the TD fixed-point, and only show that the final iterate is in the vicinity of the fixed-point defined by the amount of error accumulated over the trajectory.

In contrast, we actually proved exact convergence to the TD fixed-point by showing that each iteration is indeed a contraction irregardless of the value of $K$. Even though Lee and He are correct in saying that using a finite $K$ results in approximately solving each iteration, we can still show that each iteration remains a contraction by looking at the net effect of the $K$ updates to the online network and the single update to the target network. We prove that the net effect of these updates is one that ensures that the iterate makes steady progress towards the unique fixed-point irregardless of $K$. To the best of our knowledge, this result is completely novel, and takes a major step in supporting the soundness of TD as well as the successor algorithms (such as DQN) that use a frozen target network.

2- We believe to be the first paper to show convergence of TD in the most natural extension of the quadratic objective. To ensure that this extension is possible, we made two additional assumptions, namely the Lipschitz continuity of the objective with respect to the target network, and the strong convexity of the objective with respect to the online network. While these assumptions hold in the linear case with quadratic loss, some of the reviewers asked us to elaborate further in terms of the validity of these assumptions for mainstream loss functions and function approximators.

To this end, we present two families of loss functions where our assumptions can hold easily. In particular, to explain the first family, recall that TD could be thought of as solving for a sequence of optimization problems as follows:

$\theta^{t+1} \leftarrow \arg\min_{w} H(\theta^{t},w).$

Now suppose we can write the function $H(\theta,w)$ as the sum of two separate functions $H(\theta,w) = G(\theta, w ) + L(w)$, where the function $L(w)$ is strongly convex with respect to $w$. This setting is akin to using ridge regularization which is quite common in deep learning (for example AdamW). This allows us to now work with functions $G$ that are only convex (in fact they technically can be weakly convex) with respect to $w$. We provide two examples:

1- Suppose we would like to stick with the linear function approximation architecture. Then, the function $G$ could be constructed using any convex loss where $\nabla_w G(\theta,w)$ is Lipschitz-continuous with respect to $\theta$. Examples that satisfy this include the Logistic loss or the Huber loss.

2- Suppose we want to use the more powerful convex neural networks. We need the loss function to be convex and  monotonically increasing so that the resultant function $G$ is still convex. This is due to the classical result on the composition of convex operators. One example is the quadratic loss where we restrict the output of the function approximator to positive values. Such neural nets are also Lipschitz continuous given proper activation functions such as ReLU.

Beyond this family, since the submission we have identified a second family, namely the control setting (beyond prediction) where a greedification operator is needed for bootstrapping. For example, with the quadratic loss we could have:

$H(\theta,w)=\frac{1}{2}\sum_sd(s)\sum_a\pi(a|s)(E_{s'}[r+ \gamma\max_{a'}q(s',a',\theta)]-q(s,a,w))^2.$

We again need the two assumptions, namely strong-convexity with respect to $w$ and Lipschitzness of $\nabla_w H(\theta,w)$ with respect to $\theta$ to hold. Actually, Lee and He (2020) already showed the strong convexity of the objective with respect to $w$, but we need to still show the Lipschitz property of $\nabla_w H(\theta,w)$ with respect to $\theta$. Note that Lee and He (2020) showed the Lipschitz property only with respect to $w$ and not with respect to $\theta$. We are now able to show this result. Please see the proof in the pdf attached to this rebuttal. Our proof also supports other greedification operators, such as softmax, so long as these operators are non-expansive. We will add this result to the paper. So, together, this would be another example of the kind of loss functions of the form $H(\theta,w)$ that satisfy our assumptions.

Lee and He ``Target-based Q-learning", 2020.

---

### Decision · Program_Chairs · 2023-09-21

**Decision:**

Accept (poster)

**Comment:**

This paper studies the convergence of TD algorithms by viewing TD as iterative optimization procedures. The authors identified "optimization force" and "target force" as determining factors for convergence or divergence, and the conclusion is that TD converges when the optimization forces overcomes the target force.

The viewpoint of optimization and target forces seems novel to most reviewers and the AC, and it makes good progresses toward better understanding convergence of TD methods. Reviewers found that the paper is well-written and easy to follow.

There were several questions regarding comparing to existing results and weakening the strong convexity assumptions from the reviews that the authors provided responses during rebuttal. Please include these clarifications and discussions in revisions to strengthen the current exposition.